# Persistent TFIIH binding to non-excised DNA damage causes cell and developmental failure

Alba Muniesa-Vargas[1,3], Carlota Davó-Martínez [1,3], Cristina Ribeiro-Silva [1,3], Melanie van der Woude[1], Karen L. Thijssen [1], Ben Haspels[2], David Häckes[1], Ülkem U. Kaynak [1], Roland Kanaar[2], Jurgen A. Marteijn [2], Arjan F. Theil [1], Maayke M. P. Kuijten[2], Wim Vermeulen [1] & Hannes Lans [1]✉

Congenital nucleotide excision repair (NER) deficiency gives rise to several cancer-prone and/or progeroid disorders. It is not understood how defects in the same DNA repair pathway cause different disease features and severity. Here, we show that the absence of functional ERCC1-XPF or XPG endonucleases leads to stable and prolonged binding of the transcription/DNA repair factor TFIIH to DNA damage, which correlates with disease severity and induces senescence features in human cells. In vivo, in *C. elegans*, this prolonged TFIIH binding to non-excised DNA damage causes developmental arrest and neuronal dysfunction, in a manner dependent on transcription-coupled NER. NER factors XPA and TTDA both promote stable TFIIH DNA binding and their depletion therefore suppresses these severe phenotypical consequences. These results identify stalled NER intermediates as pathogenic to cell functionality and organismal development, which can in part explain why mutations in XPF or XPG cause different disease features than mutations in XPA or TTDA.

Nucleotide excision repair (NER) is a major DNA repair pathway that promotes genome stability by removing a wide range of different lesions that distort the DNA helical structure[1,2]. Lesions repaired by NER arise from exposure to different genotoxic agents encountered in daily life, including metabolism-derived and environmental reactive chemicals and UV light from the sun. The core NER pathway is well characterized and can be summarized in four sequential steps: 1) lesion detection, 2) DNA unwinding and damage verification, 3) excision of a single-stranded DNA stretch containing the lesion and 4) DNA synthesis and ligation to restore the gap. Lesions are detected throughout the genome, by the concerted activity of the CRL4[DDB2] and XPC-RAD23B-CETN2 complexes, in a subpathway termed global genome NER (GG-NER)[3,4]. Alternatively, lesions can also be detected through transcription-coupled NER (TC-NER), where obstructed transcriptional elongation by RNA polymerase II (Pol II) leads to the stable binding of the DNA translocase CSB to lesion-stalled Pol II and recruitment of the CRL4[CSA] and UVSSA-USP7 complexes[5,6].

Following lesion detection, both subpathways converge to the same central NER machinery, starting with the recruitment of the Transcription Factor IIH (TFIIH) complex[7]. TFIIH consists of a 7-subunit core complex, comprised of ERCC2/XPD, ERCC3/XPB, GTF2H1/p62, GTF2H2/p44, GTF2H3/p34, GTF2H4/p52 and GTF2H5/TTDA, and a 3-subunit CDK-activating kinase (CAK) subcomplex. Recruitment of TFIIH to DNA damage is mediated by an interaction between p62 and XPB with either XPC, in GG-NER, or UVSSA, in TC-NER[8–11]. The arrival of TFIIH stabilizes the binding of XPC to DNA and coincides with CRL4[DDB2] dissociation[12]. DNA double helix opening by ATPases XPB and XPD, whose 5′–3′ helicase/translocase activity is blocked by DNA damage,

[1]Department of Molecular Genetics, Erasmus MC Cancer Institute, Erasmus University Medical Center, 3015 GD Rotterdam, The Netherlands. [2]Department of Molecular Genetics, Erasmus MC Cancer Institute, Oncode Institute, Erasmus University Medical Center, 3015 GD Rotterdam, The Netherlands. [3]These authors contributed equally: Alba Muniesa-Vargas, Carlota Davó-Martínez, Cristina Ribeiro-Silva. ✉e-mail: w.lans@erasmusmc.nl

serves to scan for the presence of a lesion and creates a suitable bubble substrate for subsequent endonucleolytic incision[13–15]. Together with TFIIH, XPA and RPA are recruited, which, based on in vitro assays, are thought to interact with and coordinate DNA incision by the ERCC1-XPF and XPG endonucleases[16–18]. Furthermore, XPA stimulates XPD helicase activity in vitro by promoting the release of the CAK subcomplex from TFIIH and may thus also serve as an auxiliary factor in damage verification[15,19,20]. ERCC1-XPF and XPG coordinately incise the DNA 5' and 3' from the lesion once a proper DNA substrate is formed through the stable binding of the NER incision complex consisting of TFIIH, RPA, XPA, ERCC1-XPF and XPG[21–23]. This is followed by the dissociation of TFIIH and other incision complex factors with help from the HLTF translocase[24]. The resulting single-stranded DNA gap is finally restored by de novo DNA synthesis and ligation[25].

Hereditary NER defects cause several rare autosomal recessive pleiotropic diseases with widely varying phenotypic expression[26]. Mutations in genes *XPA* to *XPG* can cause xeroderma pigmentosum (XP), whereas mutations in *CSA* and *CSB* can cause Cockayne syndrome (CS) or, together with mutations in *UVSSA*, UV-sensitive syndrome. XP is characterized by photosensitivity, pigmentation abnormalities, and a very high risk of developing cancer[27]. Additionally, some XP patients develop progressive neurodegeneration. CS, on the other hand, is characterized by severe growth failure, progressive neurodegeneration, and segmental progeroid features but not by cancer predisposition[28]. UV-sensitive syndrome is characterized by mild photosensitivity of the skin. Mutations in some genes of the NER incision complex, i.e., in *XPB*, *XPD*, *XPF* and *XPG*, can cause a syndrome with combined features of XP and CS referred to as xeroderma pigmentosum-Cockayne syndrome (XPCS) complex, or, in its most severe form, as cerebro-oculo-facio-skeletal syndrome (COFS)[29,30]. Moreover, other mutations in TFIIH genes *XPB*, *XPD* and *TTDA* can cause a photosensitive form of the Trichothiodystrophy (TTD) syndrome, which is characterized by brittle hair and nails, developmental delay and cognitive decline[31,32]. Typically, mutations in *XPA* never give rise to XPCS, although they can cause a severe form of XP combined with growth failure and progressive neurodegeneration that is sometimes referred to as DeSanctis Cacchione (DSC) syndrome[33]. It is not entirely clear why mutations in the same DNA repair pathway, and even in the same gene, cause the different symptoms associated with each different disorder, which range in severity from mild to very severe (with mortality at young age). It is thought that symptoms affecting terminally differentiated tissues in TTD patients are derived from gene expression problems due to TFIIH instability[31] and that cancer predisposition in XP is caused by mutation accumulation due to reduced GG-NER activity[27]. However, the pathogenesis of other symptoms, in particularly those associated with CS and affecting the nervous system, likely entails additional mechanisms. Indeed, besides NER deficiency also defects in other DNA repair pathways, transcription, stress responses and/or mitochondria have been linked to CS[5,34–36].

We previously observed that the central NER machinery is continuously targeted to DNA damage in cells harboring XPCS-causative XPF mutations and postulated that accumulation of DNA repair intermediates may interfere with transcription and/or replication and could be causative for CS-related neurodegenerative symptoms[5,37]. Here, we aimed to further clarify the molecular pathology underlying XPCS. We studied differences between cells lacking XPA or TTDA and cells lacking XPF or XPG, which could help explain why mutations in the NER incision complex factors XPA and TTDA are never associated with CS features. We show that both XPA and TTDA promote the stable binding of TFIIH to DNA damage, whereas ERCC1-XPF and XPG nucleolytic activity promotes its dissociation. As a result, TFIIH is stably and more persistently bound to DNA damage in cells without proper DNA damage excision. Our results suggest that this persistent TFIIH binding induces senescence features in cultured cells and, as we

show using *C. elegans* as model system[38], leads to TC-NER-dependent functional impairment of differentiated neurons in vivo.

## Results

### XPA and XPF/XPG differentially influence TFIIH binding to DNA damage

To investigate the mechanistic differences in NER between cells lacking XPA and cells lacking ERCC1-XPF or XPG function, we determined how the loss of these NER factors affects TFIIH association with DNA damage. To this end, we made use of previously generated GFP-XPB MRC-5 knock-in (KI) fibroblasts, in which a GFP tag is fused to the N-terminus of the endogenous TFIIH subunit XPB[12]. The dynamic cellular properties of the endogenous TFIIH complex, and how these change in response to UV radiation, were determined by fluorescence recovery after photobleaching (FRAP) experiments. In FRAP, incomplete fluorescence recovery in a region of interest indicates that a fraction of GFP-tagged factors is immobilized, which in the case of TFIIH after UV-irradiation reflects its binding to DNA damage during NER[12,39,40]. In cells treated with control siRNA (siCTRL), a fraction of TFIIH indeed became immobilized immediately after UV irradiation, indicating that a significant fraction of TFIIH molecules was active in NER (Fig. 1A–C). The fraction of immobilized TFIIH was decreased when FRAP experiments were conducted 3 h after UV irradiation, because of ongoing repair of the DNA damage (Supplementary Fig. 1A; Fig. 1C). However, in cells in which XPF or XPG were depleted (Supplementary Fig. 1B), TFIIH showed a strikingly larger UV-induced immobile fraction, which was still observed 3 h after UV irradiation. These results indicate that in the absence of DNA damage excision TFIIH binds longer to DNA damage. In sharp contrast, XPA depletion led to a smaller UV-dependent immobile fraction of TFIIH than the control condition, both immediately and 3 h after UV irradiation (Fig. 1B, C; Supplementary Fig. 1B, C). Altogether, these data indicate that XPA stabilizes TFIIH binding to DNA damage and that XPF and XPG stimulate TFIIH dissociation.

### XPA promotes the formation of a stable NER incision complex

The UV-induced immobile fraction of TFIIH in cells with XPA depletion diminished in time, similar to control siRNA-treated cells, indicating that residual repair likely still takes place due to incomplete XPA knockdown. We, therefore, generated cells with XPA or XPF knockout (KO) alleles expressing fluorescent TFIIH, to corroborate our RNAi-based results and to test if the strong UV-induced retention of TFIIH in UV-treated XPF deficient cells is XPA dependent. We knocked out XPA in the GFP-XPB KI MRC-5 cell line (Supplementary Fig. 1D) and knocked in GFP at the endogenous XPB locus in wild-type U2OS cells and in previously generated XPF KO U2OS cells[37] (Supplementary Fig. 1E, F).

Similar to siRNA-mediated XPA depletion, the fraction of TFIIH molecules bound to DNA damage immediately after UV exposure was lower in XPA KO cells than in XPA proficient (or WT) cells (Fig. 1D-F). Moreover, also 3 h after UV irradiation showed a higher percentage of TFIIH molecules bound to DNA in the XPA KO cells, likely due to repeated binding in the complete absence of repair in these cells. Strikingly, depletion of XPF in XPA KO cells did not lead to increased TFIIH binding to chromatin (Fig. 1E, F), contrary to the strong GFP-XPB immobilization in wild-type MRC-5 cells after XPF depletion (Fig. 1A, C). FRAP in wild-type U2OS cells also showed a clear UV-induced immobilization of GFP-XPB, which was reduced upon XPA depletion but enhanced upon XPF KO (Fig. 1G–I). Interestingly, the enhanced GFP-XPB immobilization in XPF KO cells was prevented by XPA depletion. These data indicate that the prolonged stable binding of TFIIH in the absence of DNA damage excision depends on XPA, and that XPA is crucial for the formation of a stable incision complex. Consequently, XPF deficiency does not influence TFIIH binding to DNA damage in XPA deficient cells.

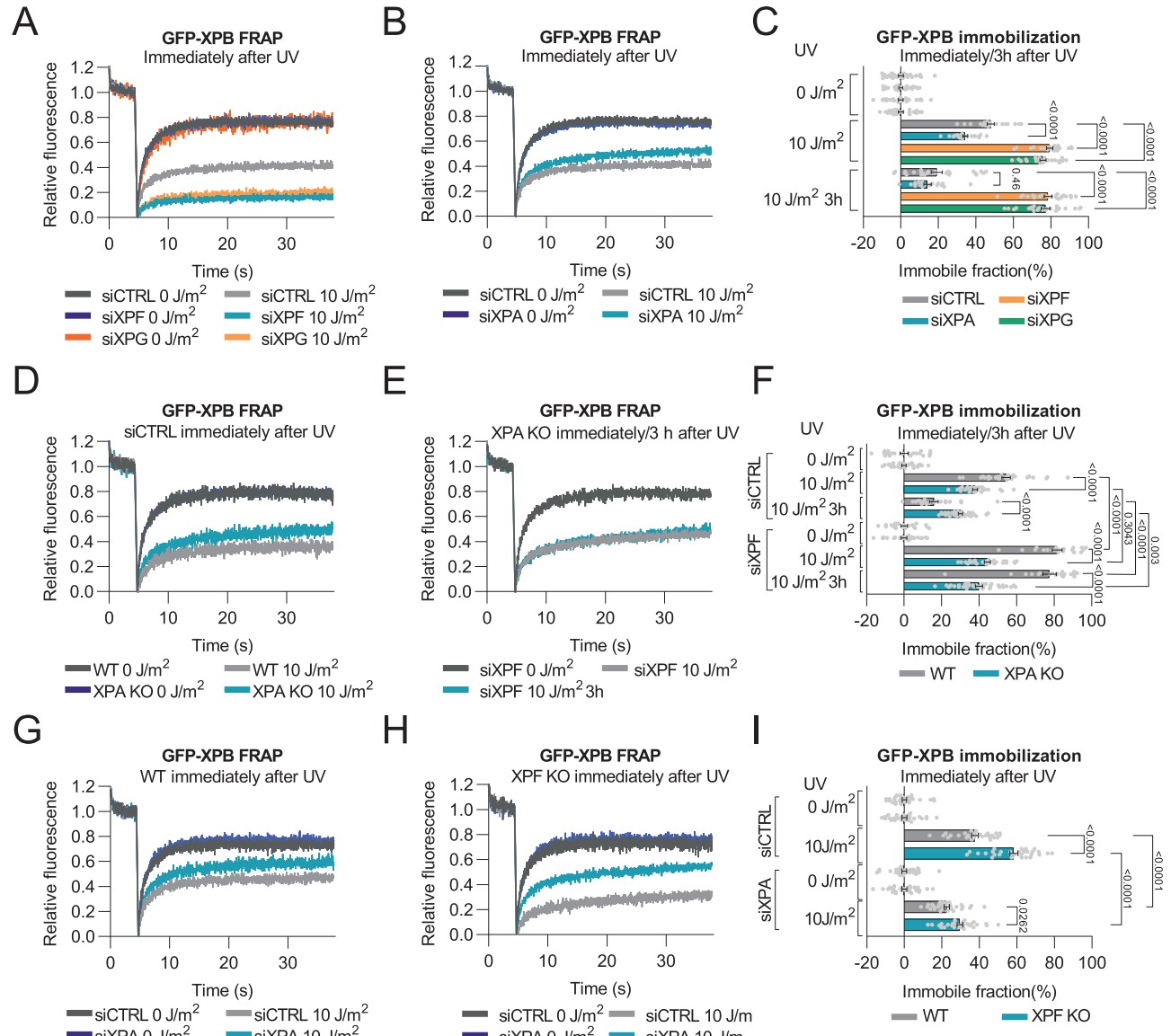

**Fig. 1 | TFIIH-DNA damage binding differs between XPA and XPF or XPG deficient cells.** **A**–**C** FRAP of GFP-tagged XPB and quantification of immobile fraction in MRC-5 cells treated with control (CTRL), XPF, XPG or XPA siRNA, non-irradiated (0 J/m²), immediately after (10 J/m²) or 3 h after (10 J/m² 3 h; FRAP curves shown in Supplementary Fig. 1A, C) UV-C irradiation. $n = 29, 22, 26$ (for siCTRL 0, 10 J/m² and 3 h 10 J/m²); $n = 30, 22, 23$ (for siXPF 0, 10 J/m² and 3 h 10 J/m²); $n = 22, 24, 27$ (for siXPG 0, 10 J/m² and 3 h 10 J/m²); $n = 30, 22, 19$ (for siXPA, 10 J/m² and 3 h 10 J/m²). **D**–**F** FRAP of GFP-tagged XPB and quantification of immobile fraction in wild-type (WT) and XPA knockout (KO) MRC-5 cells treated with control (CTRL) or XPF siRNA, non-irradiated (0 J/m²), immediately after (10 J/m²) or 3 h after (10 J/m² 3 h) UV-C irradiation. $n = 20, 30, 22$ (for WT siCTRL 0, 10 J/m2 and 3 h 10 J/m²);

$n = 20, 20, 20$ (for WT siXPF 0, 10 J/m² and 3 h 10 J/m²); $n = 30, 30, 33$ (for XPA KO siCTRL 0, 10 J/m² and 3 h 10 J/m²); $n = 30, 30, 34$ (for XPA KO siXPF 0, 10 J/m² and 3 h 10 J/m²). **G**–**I** FRAP of GFP-tagged XPB and quantification of immobile fraction in wild-type (WT) and XPF knockout (KO) U2OS cells treated with control (CTRL) or XPA siRNA, non-irradiated (0 J/m²) or immediately after 10 J/m² UV-C irradiation. $n = 30, 30$ (for WT siCTRL 0 and 10 J/m2); $n = 30, 30$ (for WT siXPA 0 and 10 J/m2); $n = 30, 30$ (for XPF KO siCTRL 0 and 10 J/m2); $n = 30, 30$ (for XPF KO siXPA 0 and 10 J/m2). Curves are normalized to bleach depth. Each FRAP curve represents the average of at least three independent experiments. Numbers represent $p$-values (ANOVA corrected for multiple comparisons). Error bars represent SEM. Source data are provided as a Source Data file.

## XPA deficiency prevents TFIIH stalling in excision deficient cells

To independently substantiate these FRAP results, we assessed the loading of endogenous non-tagged TFIIH onto damaged chromatin in wild-type and XPF KO U2OS cells, in the presence and absence of XPA, at an early (30 min) and late (3 h) time point after UV irradiation by cell fractionation. Both XPB and XPD TFIIH subunits showed an increased and continuous binding to damaged chromatin in XPF KO cells but not in wild-type cells (Fig. 2A). XPA depletion reduced the binding of TFIIH to damaged chromatin in both wild-type and XPF KO cells, in accordance with the FRAP data, and prevented continuous TFIIH binding in XPF KO cells.

To further corroborate these findings, we compared the dissociation of TFIIH from DNA damage by inverse fluorescence recovery after photobleaching (iFRAP). To this end, using a microporous filter we locally UV-irradiated the GFP-XPB KI wild-type U2OS cells and GFP-XPB KI XPF KO U2OS cells to induce steady-state accumulation of GFP-XPB at local UV damage (LUD). Subsequently, the nuclear fluorescent signal outside the LUD was continuously bleached, and the fluorescence decay over time in the LUD was measured to assess TFIIH residence time at damaged DNA (Fig. 2B). GFP-XPB dissociation in XPF KO cells was slower compared to that in wild-type cells, showing that TFIIH resides significantly longer at DNA damage in the absence of an

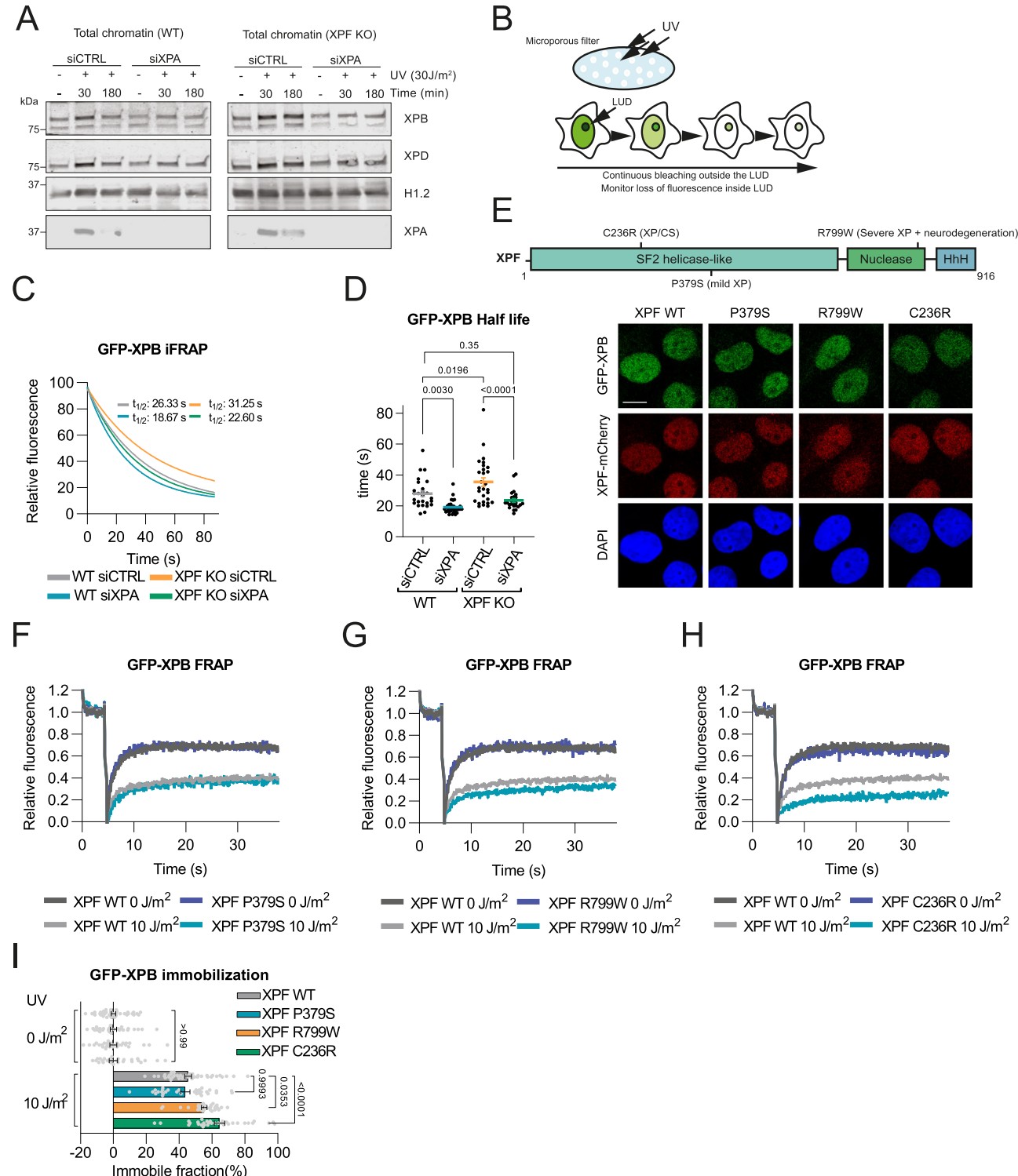

**Fig. 2 | Prolonged TFIIH binding to damaged chromatin in XPF deficient cells is prevented by XPA loss. A** Chromatin fraction after cell fractionation of WT and XPF KO U2OS cells treated with CTRL and XPA siRNAs analyzed by immunoblot against XPB, XPD, XPA and histone H1.2 (loading control). Cells were non-irradiated (-) or irradiated with 30 J/m² UV-C and lysed after 30 or 180 min. Representative experiment is shown, which was independently replicated with similar results. **B** Scheme of inverse FRAP (iFRAP). After induction of local UV damage (LUD), fluorescent signal outside LUD is continuously bleached and the decay of fluorescence in the LUD is measured over time. **C** iFRAP reflecting GFP-XPB dissociation from LUD in WT and XPF KO U2OS cells, treated with siCTRL or siXPA. Each curve is normalized to fluorescence intensity before bleaching and represents mean of three independent experiments. **D** Half-life of GFP-XPB in the LUD calculated from iFRAP in (**C**). Numbers represent p-values (ANOVA corrected for multiple comparisons). Mean and SEM of

three independent experiments. n = 26 for WT siCTRL, n = 32 for WT siXPA, n = 29 for XPF KO siCTRL, n = 24 for XPF KO siXPA. **E** Upper panel: XPF schematic structure with C236R, P379S and R799W mutations. Lower panel: Representative images of fixed GFP-XPB KI XPF KO U2OS cell lines complemented with mCherry-tagged cDNA of wild-type (WT) XPF or XPF with P379S, R799W and C236R mutations. Scale bar, 10 μm. Experiment was independently replicated with similar results. **F**–**H** FRAP of GFP-XPB in XPF KO U2OS cells expressing mCherry-tagged WT XPF or XPF with P379S (**F**), R799W (**G**) and C236R (**H**) mutations, without and with UV-irradiation (10 J/m²). **I** Percentage of GFP-XPB immobile fraction calculated from FRAP in (**F**)–(**H**). Mean and SEM of three independent experiments. Numbers represent p-values (ANOVA corrected for multiple comparisons). n = 40, 42 (for XPF WT 0 and 10 J/m²), n = 28, 31 (for XPF P379S 0 and 10 J/m²), n = 30, 30 (for XPF R799W 0 and 10 J/m²) and n = 31, 34 (for XPF C236R 0 and 10 J/m²). Source data are provided as a Source Data file.

incision (Fig. 2C, D). Conversely, GFP-XPB dissociation was faster upon XPA depletion, both in XPF proficient cells and in XPF KO cells, showing that TFIIH resides shorter at DNA damage in the absence of XPA. Together, these data confirm that XPA promotes the stable binding of TFIIH to damaged chromatin and demonstrate that the increased UV-induced TFIIH immobilization in excision deficient cells observed with FRAP is due to prolonged TFIIH binding to DNA damage.

## TFIIH stalling correlates with the severity of cellular impairment and NER disease symptoms

Given the difference of TFIIH-DNA binding kinetics to DNA damage in XPA compared to XPF and XPG deficient cells, we wondered if prolonged TFIIH binding impairs cell function. Therefore, we first tested if the degree of TFIIH stalling correlates with the severity of NER diseases caused by different XPF mutations. We stably transduced mCherry-tagged wild-type XPF (WT) and three disease-associated mutant XPF cDNA constructs, encoding P379S mutant XPF, associated with mild XP, R799W mutant XPF, associated with more severe XP including progressive neurodegeneration, and C236R mutant XPF, associated with the most severe XPCS complex, in GFP-XPB KI XPF KO U2OS cell lines (Fig. 2E)[37,41–43]. FRAP analysis of the mobility of GFP-XPB after UV irradiation showed that expression of both WT and P379S mutant XPF led to a similar level of TFIIH association with damaged chromatin (Fig. 2F). However, expression of R799W mutant XPF caused increased TFIIH binding to damaged DNA (Fig. 2G), and TFIIH-DNA binding was even more pronounced in cells that expressed the C236R mutant XPF construct (Fig. 2H, I). Previous in vitro incision assays showed that C236R, but not R799W, strongly reduces XPF's nucleolytic activity[42,44]. These results, therefore, reveal that disease-associated mutations in XPF that differ in severity and catalytic activity, also differentially impact TFIIH binding to DNA damage, and that, intriguingly, the degree of TFIIH stalling correlates with the severity of the disease.

Subsequently, we investigated if cellular phenotypes due to ERCC1-XPF or XPG deficiency can be suppressed by additional XPA deficiency, thus preventing TFIIH stalling. We first determined if XPA and XPF deficiency differentially affect DNA damage-induced transcription inhibition. As loss of XPA and XPF impairs the repair of transcription-blocking DNA lesions, this also limits the recovery of RNA synthesis after UV irradiation. We knocked out XPA in wild-type and XPF KO U2OS cells by CRISPR/Cas9 (Supplementary Fig. 1G) and measured recovery of RNA synthesis by fluorescently visualizing 5-ethynyl-uridine incorporation into nascent RNA[45]. Strikingly, this showed that XPF KO leads to a slightly stronger recovery of RNA synthesis defect than XPA KO and that XPA KO suppressed this stronger defect (Fig. 3A). In contrast, we confirmed that XPA, XPF or XPG KO or double XPA and XPF KO equally leads to a complete impairment of NER activity, as shown by the complete absence of 5-ethynyl-2′-deoxyuridine incorporation into DNA, as measured by Unscheduled DNA Synthesis[45], after UV irradiation (Supplementary Fig. 1H). Therefore, these results suggest that possibly the increased TFIIH stalling at DNA damage in XPF KO cells contributes to the stronger impairment of transcriptional activity observed in these cells. To determine if this transcriptional impairment correlates with a difference in the level of lesion-induced Pol II stalling, we performed FRAP in previously generated KI U2OS cells in which the GFP variant mClover is fused to endogenous CSB[46]. The translocase CSB stably binds to lesion-stalled Pol II. Therefore, CSB immobilization, as measured by FRAP, is a very sensitive indicator of DNA damage-induced Pol II stalling[46,47]. Indeed, we observed clear immobilization of a fraction of CSB-mClover molecules after UV irradiation, but this fraction was comparable in cells treated with control siRNA or with siRNAs targeting XPA, XPF or XPG (Fig. 3B, C). These results suggest that increased transcriptional impairment due to XPF deficiency, compared to XPA deficiency, is not due to differences in Pol II stalling.

Endogenous DNA damage accumulation due to ERCC1-XPF or XPG deficiency, but not XPA deficiency, causes cellular senescence in mouse models and expression of senescence-associated secretory phenotype factors such as IL6 in human fibroblasts. These DNA damage-induced senescence features are hypothesized to contribute to the progeroid features observed in NER disease[48–53]. Therefore, we tested if XPA and XPG deficiency differentially affect endogenous DNA damage-induced changes in the expression of nuclear lamins, which is a biomarker for senescence induction[48,54], and in the expression of IL6, whose upregulation is associated with senescence induction[55]. To measure changes in these markers, we stably expressed a lamin- and IL6-expression-based reporter system in wild-type and XPG KO U2OS cells (Supplementary Fig. 2A). This system encompasses translational reporters to measure changes in the expression and expression ratio of BFP-tagged lamin B1 and mScarlet-tagged lamin A, and a transcriptional reporter to measure changes in GFP expression driven by the IL6 promoter. The functionality of this system was confirmed by treating wild-type cells with doxorubicin, which is a genotoxic drug known to induce senescence features[56]. Doxorubicin treatment caused a clear change in the expression and expression ratio of lamin B1 and lamin A, which we visualized by plotting the lamin B1 and lamin A fluorescence intensities and by plotting the proportion of cells with deviating lamin B1 and lamin A expression (Supplementary Fig. 2B–E). Doxorubicin treatment also led to an increased IL6 expression in cells, shown in the same graphs with a blue color to indicate IL-6 positive cells (Supplementary Fig. 2B and E). In XPG KO cells, we also observed clear changes in the expression of lamin B1 and lamin A and an induction of IL6 as compared to wild-type cells, which were more pronounced as compared to siXPA-treated cells (Figure 3Di, ii, iii; Supplementary Fig. 2E–I). Furthermore, XPA depletion reduced the perturbation of lamin B1 and lamin A expression and IL6 induction in XPG KO cells (Figure 3Div, E; Supplementary Fig. 2E–I). Together, these results suggest that preventing prolonged TFIIH binding by XPA deficiency effectively suppresses cellular defects and phenotypes caused by XPF or XPG deficiency.

## XPA-1 deficiency suppresses neural dysfunction in *C. elegans* *xpf-1* and *xpg-1* mutants

To investigate if TFIIH stalling similarly impairs cell function in vivo, we studied TFIIH DNA damage recruitment and neuron functionality in XPF, XPG and XPA deficient *C. elegans* [57–60]. Previously, we generated KI animals with an auxin-inducible degradation (AID) tag[61] and GFP fused to the TFIIH subunit GTF-2H1 (GTF2H1/p62 in humans)[62], allowing us to visualize the recruitment of endogenous TFIIH to damaged DNA in vivo. TFIIH is rapidly recruited to UV-damaged chromosome pairs in oocytes of wild-type animals and dissociates within ~30 min due to efficient repair of the UV-lesions (Fig. 4A, B)[63]. Recruitment was also observed in *xpa-1* and *xpf-1* null mutant animals, but in *xpf-1* animals, this recruitment was more persistent after 30 min. Importantly, XPA-1 deficiency suppressed this persistent TFIIH recruitment in *xpf-1* animals. These results are in line with our human experimental data and indicate that XPA-1 also promotes the stable binding of TFIIH to DNA damage in *C. elegans*, while its dissociation concurs with DNA damage excision.

Functionally intact chemosensory neurons in the head of *C. elegans* can take up the fluorescent dye DiI from the environment via ciliated dendrites[64], as clearly shown by fluorescently labeled neuron bodies, dendrites, and axons in unirradiated wild-type and NER deficient animals exposed to DiI (Fig. 4C). We have previously shown that this 'dye filling' neuronal phenotype requires active transcription maintained by TC-NER and is impaired upon transcription inhibition[63]. Sensory neurons in *xpf-1* and *xpg-1* mutant animals completely lost their dye filling ability after UV irradiation, in a dose-dependent manner, indicative of transcription impairment and a loss of functional

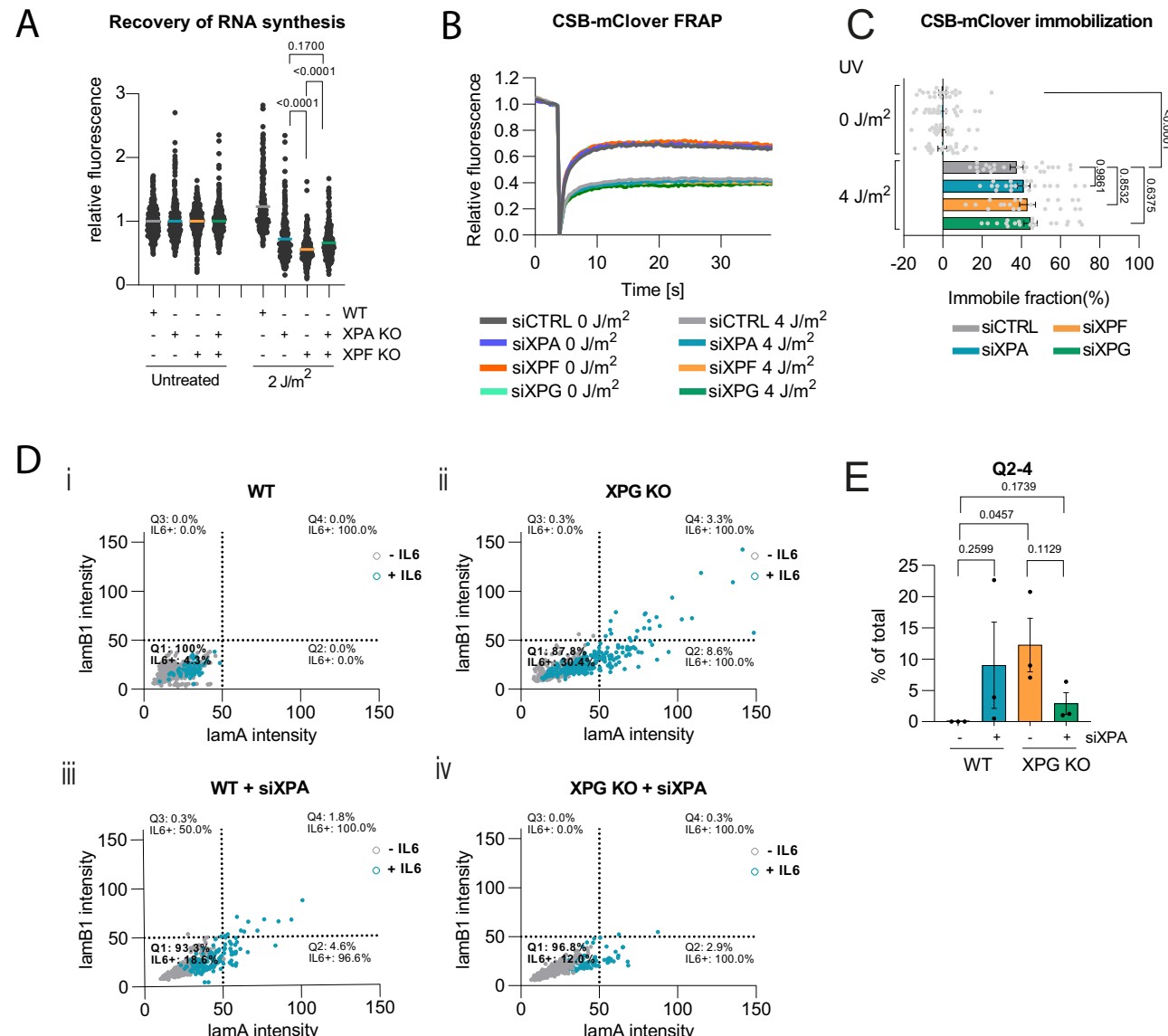

**Fig. 3 | XPA loss suppresses transcriptional and senescence features due to XPF or XPG deficiency. A** Recovery of RNA synthesis 24 h after mock-treatment (untreated) or 2 J/m² UV-C irradiation in WT, XPA KO, XPF KO and XPA/XPF KO U2OS cells. Mean with SEM of four independent experiments. Numbers represent p-values (ANOVA corrected for multiple comparisons). *n* (left to right)= 525, 507, 605, 634, 414, 333, 377 and 408. **B**, **C** FRAP of mClover-tagged CSB and quantification of immobile fraction in U2OS cells treated with control (CTRL), XPA, XPF or XPG siRNA, non-irradiated (0 J/m²) or immediately after (4 J/m²) UV-C irradiation. *n* = 20, except for siCTRL 4 J/m² *n* = 24. Shown is the average with SEM of two independent FRAP experiments. Numbers represent *p-values* (ANOVA corrected for multiple comparisons). **D** Scatter plot showing relative mScarlet-Lamin A (lamA) and BFP-Lamin B1 (lamB1) intensities in (i) wild-type (WT) (ii) XPG KO (iii) siXPA-

treated WT and (iv) siXPA-treated XPG KO U2OS cells. Each dot represents a cell. Blue color indicates senescence-associated GFP-IL6 induction. Grey color indicates no GFP-IL6 expression. Scatter plots are divided in four quadrants (Q1-Q4) and percentage of cells in each quadrant and percentage of IL6 positive cells (IL6 + ) is indicated per quadrant. Cells from three replicate experiments are shown. *n* = 2109 for WT, *n* = 666 for XPG KO, *n* = 628 for WT+siXPA and *n* = 586 for XPG KO+siXPA. Representative images of cells are shown in Supplementary Fig. 2E. **E** Graph showing the average percentage of WT or XPG KO U2OS cells, without or with siXPA treatment, in Q2, Q3 and Q4 (Q2-4). Mean with SEM from three independent replicate experiments. Numbers represent *p-values* (unpaired two-sided t-test). Source data are provided as a Source Data file.

integrity of these neurons (Fig. 4C–E). This was also observed in *xpa-1* animals, but to a lesser degree. Strikingly, loss of *xpa-1* in *xpf-1* or *xpg-1* animals rescued their severe dye filling impairment to the same level as observed in *xpa-1* single mutants (Fig. 4C–E).

We next measured developmental arrest in response to UV irradiation, as *C. elegans* larval development requires unperturbed transcription in functionally intact neurons[58,63]. Larval survival after UV-induced DNA damage can, therefore, be used as a measure of neuronal and transcriptional integrity. In the 'L1 larvae survival' assay, which measures the capacity of first stage larvae to develop into adults[65], *xpf-1* and *xpg-1* animals displayed a dose-dependent developmental arrest

upon UV irradiation, which was more pronounced than in *xpa-1* animals (Fig. 4F, G). Again, we observed that XPA-1 deficiency partially but clearly suppressed the stronger UV-induced developmental arrest of *xpf-1* and *xpg-1* animals. Taken together, these results suggest that the degree of TFIIH stalling to DNA damage directly correlates with the severity of neuronal defects observed in vivo.

**Prolonged TFIIH binding to non-excised DNA damage causes neuron dysfunction**

To confirm that prolonged TFIIH binding to DNA damage, and not some other feature of XPF or XPG deficient cells, induces severe

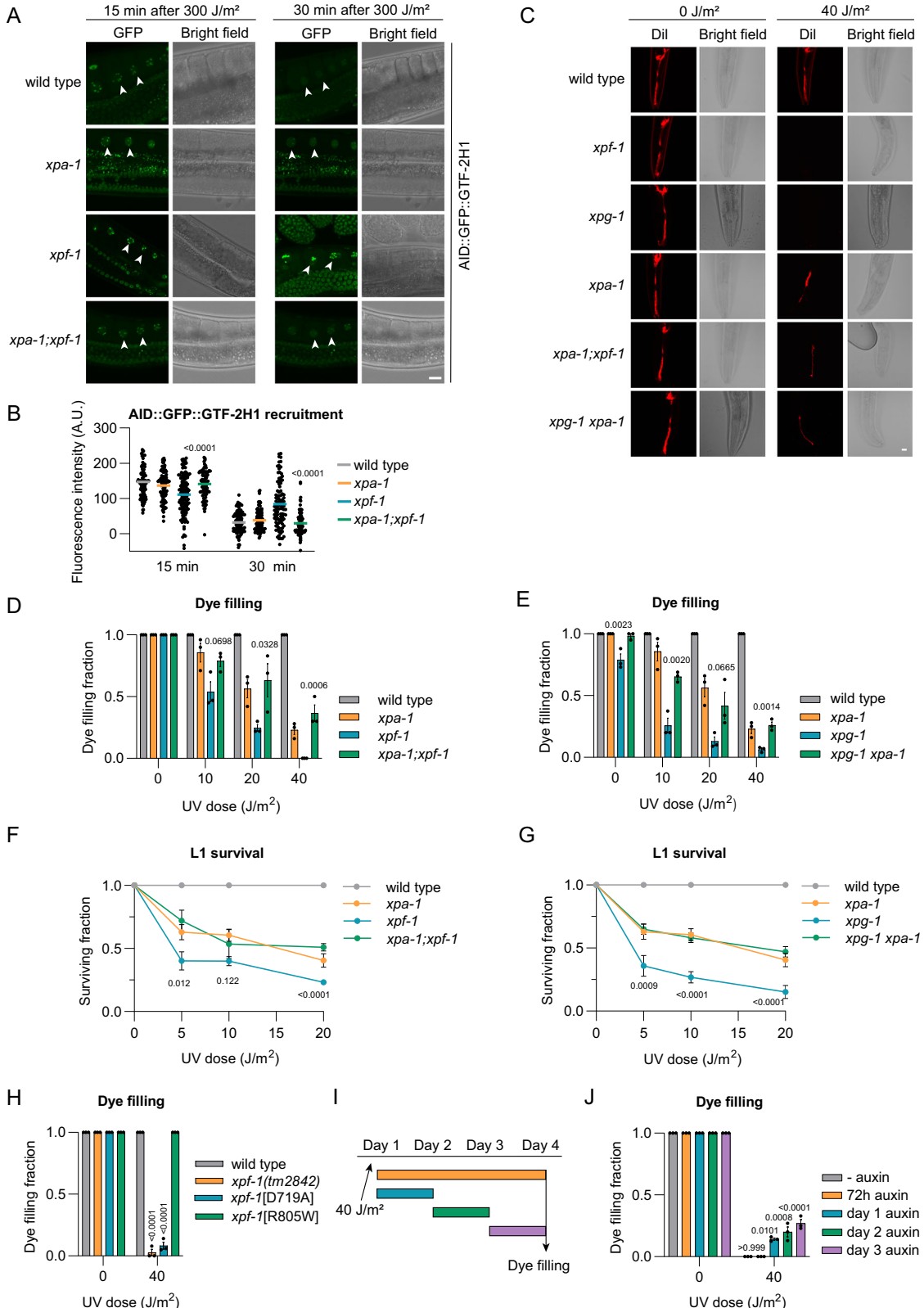

neuronal dysfunction, we tested the impact of GTF2H5/TTDA (in humans) or GTF-2H5 (in *C. elegans*) deficiency in the absence of DNA damage excision. GTF2H5/TTDA promotes the stability and DNA damage recruitment of TFIIH in human cells[66–68]. Using immuno-fluorescence and UV irradiation through a microporous filter to inflict LUD, we observed that siRNA-mediated depletion of human GTF2H5/TTDA not only suppressed TFIIH recruitment to DNA damage in wild-

type cells, but also in XPF KO U2OS cells (Supplementary Fig. 3A). In *C. elegans*, we found that GTF-2H5 also promotes the stability of the TFIIH complex[62] and that its absence prevented stable TFIIH recruitment to DNA damage as well (Supplementary Fig. 3B). Intriguingly, we observed that GTF-2H5 deficiency suppressed both the dye filling defect and the strong developmental arrest of UV-irradiated *xpf-1* animals (Supplementary Fig. 3C, D).

**Fig. 4 | XPA-dependent prolonged TFIIH-DNA damage binding impairs neuron functionality. A** AID::GFP::GTF-2H1 recruitment to oocyte chromosomes, 15 and 30 min after UV-B-irradiation of living wild-type, *xpa-1*, *xpf-1* or *xpa-1*;*xpf-1* animals. Scale bar, 10 μm. **B** AID::GFP::GTF-2H1 recruitment in fixed animals. Mean with SEM of two independent experiments. Numbers represent *p-values* (ANOVA corrected for multiple comparisons). *n* (left to right)= 110, 115, 106, 117, 129, 99, 94, 102 bivalents. **C** DiI-dye filling of wild-type, *xpf-1*, *xpg-1*, *xpa-1*, *xpa-1; xpf-1* and *xpg-1 xpa-1* animals, 72 h after UV-B irradiation. Scale bar, 10 μm. Fraction of dye filling animals 72 h after increasing UV-B dose for (**D**) wild-type, *xpf-1*, *xpa-1* and *xpa-1; xpf-1* or (**E**) wild-type, *xpg-1*, *xpa-1* and *xpg-1 xpa-1*. Mean with SEM of three independent experiments. Numbers represent *p-values* (ANOVA corrected for multiple comparisons, comparing double mutant to *xpf-1* or *xpg-1*. *n* (left to right)= (**D**) 62, 57, 55, 60, 70, 45, 49, 48, 61, 23, 45, 21, 57, 40, 38, 41; (**E**) 64, 57, 81, 70, 73, 46, 44, 62, 62, 26, 46, 59, 58, 33, 51, 52. L1 larvae survival of (**F**) wild-type, *xpf-1*, *xpa-1* and *xpa; xpf-1* or (**G**) wild-type, *xpg-1*, *xpa-1* and *xpg-1 xpa-1* animals. Mean with SEM of three

independent experiments. Numbers represent *p-values* (ANOVA corrected for multiple comparisons), comparing *xpf-1* or *xpg-1* to double mutant. *n* (left to right)= (**F**) WT:853, 865, 874, 822; *xpf-1*:351, 334, 337, 229; *xpa-1*:1326, 1671, 1564, 1027; *xpa-1*;*xpf-1*:743, 668, 702, 521; (**G**) WT:853, 865, 874, 822; *xpg-1*:238, 220, 113, 104; *xpa-1*:1326, 1671, 1564, 1027; *xpg-1 xpa-1*:1389, 1475, 1177, 752. **H** Fraction of dye filling wild-type and *xpf-1(tm2842)*, *(emc98*[D719A]) and *(emc105*[R805W]), 72 h after 40 J/m² UV-B-irradiation. Mean with SEM of three independent experiments. Numbers represent *p-values* (ANOVA corrected for multiple comparisons), compared to wild-type. *n* (left to right)= 135, 88, 150, 150, 134, 104, 137, 150. **I** Scheme of dye filling assay indicating time periods of AID::GFP::GTF-2H1 depletion. **J** Fraction of dye filling animals with AID::GFP::GTF-2H1 depleted as in (**I**). Mean with SEM of three independent experiments. Numbers represent *p-values* (ANOVA corrected for multiple comparisons). *n* (left to right)= 55, 30, 47, 52, 45, 45, 48, 31, 37, 32. Source data are provided as a Source Data file.

Subsequently, we generated a *C. elegans* mutant strain with nuclease deficient *xpf-1*, as in human cells we observed prolonged TFIIH binding to DNA damage with the impaired XPF nuclease mutant C236R. We changed aspartic acid at position 719 (D719) of the XPF-1 nuclease motif into alanine (Supplementary Fig. 3E), which in humans abolishes XPF catalytic activity[69] and leads to continuous TFIIH recruitment to DNA damage[37]. As comparison, we also generated *C. elegans* with arginine at position 805 (R805) mutated into tryptophan (Supplementary Fig. 3F), which mimics human patient XPF mutation R799W that still retains catalytic activity[44] and leads to less pronounced TFIIH recruitment and stalling (Fig. 2G, I)[37]. Dye filling experiments showed that nuclease dead *xpf-1*[D719A] mutants, but not *xpf-1*[R805W] mutants, display a profound neuronal defect after UV irradiation (Fig. 4H).

We then directly tested whether TFIIH itself is responsible for the strong neuronal defects in UV-irradiated *xpf-1* animals. We depleted TFIIH in a mutant *xpf-1* background by growing *xpf-1* animals expressing AID-tagged GTF-2H1 and a TIR1 E3 ubiquitin ligase complex in neurons in the presence of auxin (Supplementary Fig. 3G). TFIIH depletion for the entire duration of the dye filling assay did not rescue the dye filling defect in *xpf-1* animals, in line with our previous observation that transcription is required for neuronal functionality[63]. However, transient TFIIH depletion, i.e. for 24 h of the 72 h recovery time after DNA damage induction, partially rescued the *xpf-1* dye filling defect (Fig. 4I, J). Together, these results indicate that prolonged binding of TFIIH to non-excised DNA damage is, in part, responsible for impairment of neuronal functionality in vivo.

### TFIIH stalling in transcribed DNA impairs neuron function

In *C. elegans* neurons, NER is particularly active to maintain the integrity of transcribed genes rather than that of the entire genome[63], which implies that the TFIIH stalling and neuronal defects in UV-irradiated *xpf-1* animals are predominantly due to TC-NER activity. Indeed, the loss of CSB-1 (TC-NER initiation), but not of XPC-1 (GG-NER initiation), partially rescued the dye filling defect of UV-irradiated *xpf-1* animals (Fig. 5A, B). However, counterintuitively, *csb-1* animals did not show any dye filling defect by themselves. We have previously shown that in the absence of CSB-1, XPC-1-mediated NER will remove part of the DNA damage in active genes of *C. elegans* postmitotic cells[58,63]. We confirmed this by measuring the 'recovery of protein synthesis' after DNA damage induction, which reflects DNA damage repair in a transcribed *GFP* gene in *C. elegans* muscle cells[70]. In this assay, AID-tagged GFP expressed in muscle cells is depleted by growing animals in the presence of auxin. Animals are then irradiated to induce DNA damage and after 48 h the ability to transcribe GFP is determined by measuring GFP fluorescence levels. As GFP can only be expressed if the DNA damage in genes is repaired, this assay is an indirect method to monitor DNA repair activity in transcribed genes. This assay indeed showed that repair normally takes place via CSB-1, and thus TC-NER,

but that XPC-1 acts partially redundant to CSB-1 (Supplementary Fig. 3H). This suggests that in CSB-1 deficient animals, sufficient residual DNA repair can take place during the course of the dye filling experiment (72 h) to prevent neuronal defects. Also, this implies that, in CSB-1 deficient animals, some TFIIH molecules still bind to DNA damage because of XPC-1 activity. We, therefore, tested if simultaneous loss of CSB-1 and XPC-1 rescued dye filling in *xpf-1* animals, but found that this by itself already caused a strong dye filling defect irrespective of whether animals were proficient or deficient in XPF-1 (Fig. 5C). Therefore, to further confirm that TC-NER activity induces the strong neuronal defects in UV-irradiated *xpf-1* animals, we exposed animals to the TC-NER-specific DNA damaging agent illudin-S, which creates DNA damage that is not recognized by XPC[71]. Similar to UV irradiation, *xpf-1* animals exposed to illudin-S displayed a strong developmental arrest and dye filling defect, and these phenotypes were partially rescued by XPA-1 or GTF-2H5 deficiency, respectively (Fig. 5D, E). These results demonstrate that prolonged TFIIH binding to non-excised DNA damage, mediated by NER in actively transcribed DNA, is in part responsible for neuronal functional impairment in the absence of efficient DNA damage excision. Moreover, the lack of dye filling upon loss of both CSB-1 and XPC-1 suggests that similar neuronal defects occur if DNA damage detection and initiation of NER do not take place in active genes.

## Discussion

In this study, we show that the binding of TFIIH to DNA damage differs between cells that lack XPA or TTDA and cells that lack ERCC1-XPF or XPG, even though all these cells are defective in the NER pathway. The absence of ERCC1-XPF- or XPG-mediated DNA incision delays TFIIH dissociation, resulting in its stalling at the lesion site. Contrarily, XPA and TTDA loss prevents stable, and thus prolonged, TFIIH binding to DNA damage. These results imply that XPA and TTDA are both necessary for the formation of a stable NER incision complex that normally disassembles upon DNA damage excision. Furthermore, we show that the degree of TFIIH stalling is associated with the severity of XPF-deficiency disease and that TFIIH stalling negatively impacts cell function. We therefore propose that persistent binding of TFIIH to DNA damage could be a causative factor contributing to a more severe pathology observed in some ERCC1-XPF and XPG deficient patients (Fig. 5F).

After DNA damage recognition, TFIIH is recruited to DNA damage in GG-NER by XPC and in TC-NER by UVSSA[7–11,72]. In vitro NER reconstitution assays, together with immunoprecipitation and immuno-fluorescence studies, have clearly indicated that XPA and RPA are subsequently recruited in a manner dependent on TFIIH function, and aid in the recruitment of additional factors while stabilizing the incision complex[20,68,73–76]. However, our TFIIH mobility data in living cells show reduced TFIIH association with and accelerated TFIIH dissociation from damaged chromatin in XPA deficient cells. Thus, it is likely

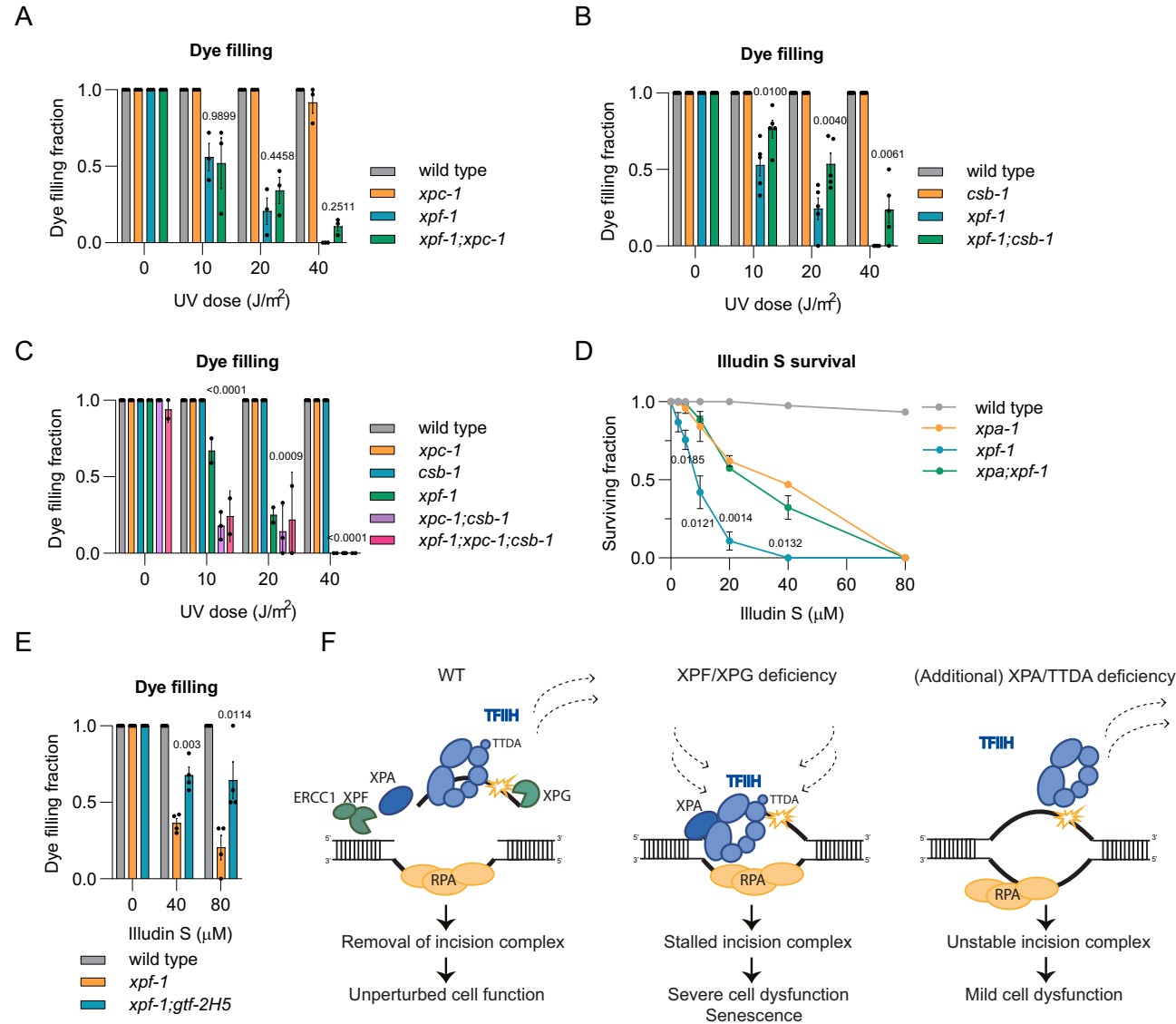

**Fig. 5 | Prolonged TFIIH-DNA damage binding due to TC-NER activity impairs neuron functionality.** Fraction of dye filling animals 72 h after increasing UV dose for (**A**) wild-type, *xpc-1*, *xpf-1*, and *xpf-1;xpc-1*, **B** wild-type, *csb-1*, *xpf-1* and *xpf-1;csb-1*, **C** wild-type, *xpc-1*, *csb-1*, *xpf-1*, *xpc-1;csb-1* and *xpf-1;xpc-1;csb-1*. Mean with SEM of two (*xpf-1;xpc-1;csb-1*) or three independent experiments. *n* (left to right)= (**A**) 75, 86, 63, 66, 81, 73, 51, 69, 73, 74, 35, 55, 75, 77, 28, 65, **B** 77, 70, 69, 67, 54, 74, 48, 68, 71, 78, 33, 50, 84, 67, 36 and 42, (**C**) 65, 59, 64, 39, 60, 32, 56, 68, 54, 30, 42, 19, 57, 63, 45, 37, 17, 61, 52, 46, 57, 19, 36, 13. **D** L1 larvae survival assay after 24 h of increasing dose illudin S in wild-type, *xpf-1*, *xpa-1* and *xpa;xpf-1*. Mean with SEM of three independent experiments. *n* (left to right)= WT:900, 900, 900, 618, 427, 814, 606; *xpa-1*:900, 855, 876, 750, 713, 963, 900; *xpf-1*:957, 732, 858, 670, 467, 725, 725; *xpa-1;xpf-1*:900, 900, 606, 401, 432, 389, 900. (**E**) Fraction of dye filling animals 72 h after 24 h treatment with illudin S for wild-type, *xpf-1* and *xpf-1;gtf-2H5*. Mean with

SEM of four independent experiments. *n* (left to right)= wild-type:72, 59, 48; *xpf-1*:68, 60, 26; *xpf-1;gtf-2H5*:66, 68, 39. Numbers in (**A**)–(**E**) represent *p-values* (ANOVA corrected for multiple comparisons), comparing the double or triple mutant to *xpf-1*. **F** Model showing difference in NER mechanism between cells with normal NER, completely lacking XPF or XPG function or lacking XPA or TTDA function. In cells with normal NER, both DNA damage and NER incision complex are removed and cells function normally. In cells completely lacking XPF or XPG, DNA damage is not removed and TFIIH stalls persistently to non-excised DNA damage, contributing to cellular dysfunction, senescence and severe NER disease. In cells lacking XPA or TTDA, DNA damage is not removed, but TFIIH does not stably bind to DNA damage, causing less cell dysfunction and a milder, or different, NER disease. Source data are provided as a Source Data file.

that XPA, in addition to its dependence on TFIIH, reciprocally promotes the stable binding of TFIIH to DNA damage. This is consistent with our previous FRAP studies showing that even the DNA association of XPC, and thus the formation of a stable DNA-bound ternary XPC-TFIIH-XPA damage verification complex, is stimulated by the presence of XPA[12]. Also, this is in line with the interdependent recruitment of TFIIH and XPA to lesion-stalled Pol II that has been observed in an in vitro TC-NER system[77]. Moreover, the similar phenotypes that we observe after the loss of TTDA and XPA fit well with this notion, as TTDA not only stabilizes TFIIH[62,66,78,79] but is also specifically required for XPA recruitment[68]. Structural, architectural modeling and

biochemical studies hint at a mechanism by which XPA stabilizes TFIIH association with DNA damage. Firstly, XPA itself interacts with DNA and TFIIH and thus anchors TFIIH to DNA damage[80]. Secondly, in doing so, XPA bridges the XPB and XPD subunits and thereby catalyzes the removal of the CAK subcomplex from TFIIH, thus stabilizing a DNA-bound TFIIH conformation that allows the XPD helicase to scan for lesions in a 5' to 3' direction[19,20,81]. Indeed, XPA stimulates XPD helicase and XPB translocase activities in vitro[14,15,19,82]. Besides XPA, XPG also stimulates the XPD helicase activity in vitro, possibly by promoting the stable binding of XPD to ssDNA and the unwinding of dsDNA at the ss/dsDNA junction[19,83]. This could imply that XPG joins TFIIH before the

full completion of DNA unwinding, suggesting that the stalling of TFIIH in XPG deficient cells is at an incomplete NER bubble intermediate.

We previously reported that severe XPCS mutations in XPF and XPG cause persistent targeting of core NER factors to DNA damage[37]. Here, we extend these findings by showing that TFIIH is not only repeatedly targeted – due to lack of repair – but also resides longer at each lesion in the absence of DNA incision. Intriguingly, we found that the degree of TFIIH binding is correlated with the severity of disease caused by XPF mutations. In particular, the XPF C236R mutation, causing severe XPCS complex[42], led to the strongest increase in TFIIH binding, which was similar to that observed upon XPF knockout. This likely relates to the fact that this mutation strongly reduces the nucleolytic activity of human XPF and almost completely abrogates XPF-mediated DNA repair activity, as measured in *Xenopus* egg extracts and human cells[37,42,84]. In line with this, we previously observed that C236R led to similar continuous DNA damage recruitment of TFIIH as a nuclease dead D715A mutation in XPF[37]. Here, we show that inactivating the nuclease activity of XPF-1 also leads to profound neuronal impairment after UV irradiation in *C. elegans*. In contrast, XPF with P379S or R799W mutations, which both cause XP, still displays considerable nucleolytic and DNA repair activity[37,44], and therefore does not lead to similarly strong TFIIH stalling as in the absence of DNA incision.

Our results indicate that longer TFIIH binding exacerbates DNA damage-induced transcription restart problems and responses consistent with cellular senescence in mammalian cells grown in culture, which can be partially suppressed by preventing stable TFIIH-DNA binding. In *C. elegans* this severely impairs the in vivo functional integrity of postmitotic, differentiated neurons upon DNA damage induction. Importantly, we have previously shown that NER is only active in transcribed DNA in these neurons and that inhibition of transcription, by depletion of XPB-1, similarly impairs neuronal functional integrity[38,58,63]. Therefore, it is likely that the problematic TFIIH stalling is due to TC-NER activity and that defects arise due to transcriptional interference. This is substantiated by the strong neuronal impairment observed after exposure of *xpf-1* animals to illudin S, which generates lesions specifically repaired by TC-NER but not recognized by GG-NER[71]. Thus, the continuous recruitment and prolonged binding of TFIIH to transcription-blocking lesions may deplete the TFIIH pool that is needed for transcription initiation and/or may lead to signaling that notifies the transcription machinery to halt transcription. Also, illegitimately stalled TFIIH could render lesions less accessible to other repair mechanisms and directly block transcription. Moreover, stalled TFIIH could, as previously observed in mouse and human XPD deficient XPCS cells, lead to extensive ssDNA formation and DNA breaks impeding transcription[85–87]. Interestingly, our results show that the complete absence of repair in transcribed genes, i.e. by mutation of both *csb-1* and *xpc-1*, leads to a dye filling defect as strong as loss of *xpf-1* or *xpg-1* in *C. elegans*. This suggests that stalling of, e.g., Pol II or the transcription machinery may impair neuronal integrity, as also previously proposed[5,88]. It will be interesting to investigate in the future which transcriptional inhibitory mechanisms underlie these similar phenotypes.

Symptoms of human patients carrying pathogenic mutations in XPA, TTDA, ERCC1-XPF or XPG are very heterogeneous and their severity depends on the type of mutation, whether the protein is still present and recruited to damage, and on life circumstances and environmental factors[26,41]. In spite of this heterogeneity, prominent differences exist between XPA or TTDA deficiency and ERCC1-XPF or XPG deficiency. Mutations that completely abrogate the XPA protein cause severe XP combined with progressive neurological decline[18,41]. However, mutations that inactivate ERCC1-XPF catalytic function or abrogate the XPG protein cause different types of neurological deficiencies, progeria, and severe developmental failure, often resulting in early death, that resemble Cockayne syndrome[23,37,42,50,89]. Such severe

features are not observed in TTDA deficient patients[67], even though the complete absence of TTDA leads to embryonic lethality in mammals[66]. These differences between XPA, ERCC1-XPF and XPG deficiency are further recapitulated in mouse models for NER disease, which show that knockout of XPA causes photosensitive features, whereas knockout of XPF or XPG leads to additional severe neurodevelopmental problems, progeria and early death[49–53,90]. These differences can be partially attributed to the different activities of these proteins, as XPA appears to be active in NER only whereas ERCC1-XPF and XPG have important functions in other genome maintenance pathways as well. For instance, ERCC1-XPF plays an important role in the unhooking of DNA interstrand crosslinks as part of the Fanconi anemia repair pathway[91,92]. Mutations in ERCC1-XPF that disrupt this function therefore lead to Fanconi anemia disease, which can include features of XP and CS pathology[42,93]. XPG has been implicated in e.g. resolving R-loops (together with ERCC1-XPF) and replication stress[94–97], which will contribute to the heterogeneous phenotype observed in XPG patients[23]. Nevertheless, our results strongly suggest that the higher degree of TFIIH stalling observed during NER, and thus the persistence of stable NER intermediates, is more toxic than the damage itself and, therefore, likely also contributes to the more severe symptoms generally observed in the absence of ERCC1-XPF or XPG (Fig. 5F). A similar hypothesis implicating stronger DNA-bound TFIIH and a long-lasting NER intermediate as causative for the more severe CS phenotype has been put forward based on analysis of Rad3/XPD mutations in *Saccharomyces cerevisiae*[98].

We show that the depletion of XPA-1 in *C. elegans*, preventing the stable association of TFIIH with DNA damage, alleviates the more severe phenotypes observed upon XPF-1 or XPG-1 deficiency. Intriguingly, XPA depletion in TC-NER deficient mouse models for (XP)CS with mutations in CSB, CSA or TFIIH, which exhibit relatively mild neurological features, severely worsens their phenotypes[86,99–103]. This was attributed to enhanced levels of endogenous DNA damage due to XPA loss and potential other DNA repair functions of CSB, CSA or TFIIH outside TC-NER. Because these findings may appear inconsistent with our results, it is important to emphasize that these mouse models cannot be directly compared to our *C. elegans* models. In *C. elegans*, we employed UV irradiation to increase DNA damage levels, as the animals live too short to accumulate sufficient endogenous DNA damage levels to manifest a clear phenotype[60]. In these conditions, *xpa-1* deficiency causes clear neurological and developmental phenotypes, as are also observed in humans with hereditary XPA deficiency[18,41], whereas XPA deficiency in mice does not cause any obvious neurological or developmental features[52,53]. Importantly, the TC-NER deficient mouse models in which XPA loss worsens the phenotype are deficient in CSB or CSA, which act upstream of TFIIH, or in TFIIH itself, and persistent TFIIH binding will therefore likely not be an issue in these animals. In contrast, we used *C. elegans* models deficient in XPF-1 or XPG-1, which both act downstream of TFIIH, and in which persistent TFIIH binding occurs. Nevertheless, it will be worth investigating in XPCS mouse models with full XPF or XPG deficiency, which display very severe neurological and growth defects[90], whether preventing the prolonged binding of TFIIH to DNA damage mitigates any of the severe features. Such experiments are necessary to explore whether it may be possible to alleviate some of the more severe symptoms of ERCC1-XPF or XPG deficiency in human patients as well, possibly by therapeutically targeting XPA or TTDA. To do this, it is necessary to determine how these NER proteins can be safely targeted and which of the heterogeneous symptoms in XPCS patients are specifically caused by TFIIH stalling.

## Methods

### Cell lines, culture conditions and treatments

All human U2OS osteosarcoma and MRC-5 fibroblasts generated and used are listed in Supplementary Table 1. Cells were cultured at 37 °C in a humidified atmosphere with 5% $CO_2$ in a 1:1 mixture of DMEM (Lonza)

and Ham's F10 (Lonza) supplemented with 10% fetal calf serum (FCS) and 1% penicillin-streptomycin (PS). All cell lines were regularly tested for mycoplasma contamination. XPA was knocked out by transient transfection of cells with pLentiCRISPR-V2[104] containing an sgRNA targeting near the START codon of the *XPA* locus (GGCGGCTTTA-GAGCAACCCG). Transfected cells were selected with puromycin and correct XPA KO clones were isolated and verified by sequencing and immunoblot. To generate GFP-XPB KI U2OS cells, cells were transiently transfected with pLentiCRISPR-v2[104] carrying an sgRNA targeting near the START codon of the *XPB/ERCC3* locus (TCTGCTGCTGTAGCTGC-CAT), and pCRBluntIITOPO carrying GFP cDNA flanked by XPB homology sequences. After selection with puromycin and FACS, a clonal cell line was isolated and verified by sequencing and immunoblot. CSB-mClover KI U2OS cells are described in Llerena Schiffmacher et al. [46]. To generate GFP-XPB KI cells expressing wild-type or P379S, R799W or C236R mutant XPF-mCherry, GFP was first knocked-in at the *XPB/ERCC3* locus in XPF KO U2OS cells[37], after which cells were lentivirally transduced with wild-type or mutant XPF-mCherry. Stable expressing cell lines were selected with either puromycin (XPF-P379S-mCherry) or blasticidin (XPF-wt-, -R799W-, -C236R-mCherry). To generate WT and XPG KO cells stably expressing BFP-Lamin B1, mScarlet-Lamin A and NLS-GFP driven by an IL6 promoter, cells were transfected with hyperactive piggyBac transposase[105] and the reporter construct containing these sequences. The reporter construct components were derived from Dronpa-LaminB1-10, mCherry-LaminA-C-18, pHRM-NLS-dCas9-GFP11x7-NLS-P2A-BFP-dWPRE and pmScarlet-i_C1. Piggybac-EGFP was a gift from Alex N. Zelensky[106] and was used as the initial cloning template. Dronpa-LaminB1-10 was a gift from Michael Davidson (Addgene plasmid # 57282); mCherry-LaminA-C-18 was a gift from Michael Davidson (Addgene plasmid # 55068); pHRM-NLS-dCas9-GFP11x7-NLS-P2A-BFP-dWPRE was a gift from Bo Huang (Addgene plasmid # 70224)[107] and pmScarlet-i_C1 was a gift from Dorus Gadella (Addgene plasmid # 85044)[108]. The plasmid with mCherry expressed under a human IL6-promotor was obtained from Tebu-bio. Cloning details are available upon request. Stable expressing cells were selected by FACS. Plasmid transfections were performed using JetPei (Promega), according to the manufacturer's instructions.

### siRNA treatments
siRNA transfections were performed 48 h before each experiment using RNAiMax (Invitrogen) according to the manufacturer's instructions and siRNA efficiency was validated by immunoblotting (Supplementary Fig. 1B). siRNA oligomers were purchased from Thermo Scientific Dharmacon: siCTRL 5′-UGGUUUACAUGUCGACUAA-3′ (D-001210-05-20), siXPA 5′ CUGAUGAUAAACACAAGCUUA-3′ (MJAWM-000011), siXPF 5′-AAGACGAGCUCACGAGUAU-3′ (D-019946-04), siXPG (M-006626-00), and siTTDA 5′-CGUCUUUGUAAUAGCAGAAUU-3′ (custom siRNA).

### Colony forming assays
For colony survivals, cells were seeded in triplicate in 6-well plates (400 cells/well) and treated with increasing doses of UV-C one day after seeding. After seven days, colonies were fixed and stained. Fixing and staining solution: 0.1% w/v Coomassie Blue (Bio-Rad) was dispersed in a 50% Methanol, 10% Acetic Acid solution. Colonies were counted with the integrated colony counter GelCount (Oxford Optronix).

### UV-C irradiation
Human cells were grown on coverslips and washed with PBS prior to UV-C irradiation with a germicidal lamp (254 nm; TUV lamp, Phillips) at the indicated doses. Local UV damage (LUD) was induced applying 60 J/m² UV-C through an 8 μm polycarbonate filter (Millipore) for immunofluorescence or 80 J/m² UV-C through a 5 μm polycarbonate filter (Millipore) for iFRAP.

### Fluorescence recovery after photobleaching (FRAP)
For FRAP, cells were seeded on coverslips and imaged with a Leica TCS SP5 or SP8 microscope using a 40x/1.25 NA HCX PL APO CS oil immersion lens (Leica Microsystems) at 37 °C and 5% CO₂. FRAP was performed as described[46,109,110]. Briefly, the nuclear fluorescent signal of the GFP-tagged protein was measured in a strip across the nucleus (512×16 pixels) at 1400 Hz of a 488 nm laser until a steady state level was reached (pre-bleach). The fluorescent signal in the strip was bleached with 100% 488 nm laser power and fluorescence recovery was measured until complete recovery. Fluorescence signals were normalized to the average pre-bleach fluorescence after background signal subtraction. The immobile fractions ($F_{imm}$) were determined using the fluorescence intensity measured immediately after UV ($I_0$) and the average steady-state fluorescent signal after complete recovery, from untreated ($I_{final, unt}$) and UV-treated cells ($I_{final, UV}$), and applying the formula:

$$F_{imm} = 1 - \frac{I_{final,UV} - I_{0,UV}}{I_{final,unt} - I_{0,UV}} \quad (1)$$

### Inverse fluorescence recovery after photobleaching (iFRAP)
For iFRAP, cells were seeded on coverslips and imaged on a Leica SP8 microscope using a 40/1.25 NA HCX PL APO CS oil immersion lens (Leica Microsystems) at 37 °C and 5% CO₂. The entire nucleus was bleached with 20% 488 nm laser power except for two non-bleached areas where fluorescence signal was monitored over time: LUD and a non-damaged nuclear area. Background signal was measured in the cytoplasm. iFRAP curves were obtained after background correction and normalization of the damaged area to the fluorescence levels of the pre-damage conditions. The non-damaged area served as internal control and was not plotted. To obtain the half-time of protein residence in the LUD the non-linear regression fitted to one-phase exponential decay analysis was applied to the iFRAP curves, using Graph Pad Prism version 9 for Windows (GraphPad Software, La Jolla California USA).

### Cell fractionation
U2OS cells were grown to confluency on 10 cm dishes, UV-C irradiated with the indicated dose and lysed in lysis buffer (30 mM HEPES pH 7.6, 1 mM MgCl₂, 130 mM NaCl, 0.5% Triton X-100, 0.5 mM DTT and EDTA-free protease inhibitor cocktail (Roche)), at 4 °C for 30 min. Non-chromatin bound proteins were recovered by centrifugation (10 min, 4 °C, 16100 g). Chromatin-containing pellet was resuspended in lysis buffer supplemented with 250 U/μL of Benzonase (Merck Millipore) and incubated for 1 h at 4 °C. Equal amounts of sample were used for SDS-PAGE gels and immunoblotting analysis.

### Immunoblotting
Total cell extracts were prepared by washing cells twice with PBS and harvesting in 1:1 mixture PBS and 2x Concentrate Laemmli buffer (Merck Sigma-Aldrich) containing 1:1000 benzonase (Millipore) and boiled for 5 min at 98 °C. Protein lysates were separated in SDS-PAGE gels and transferred onto PVDF membranes (0.45 μm, Merck Millipore). Membranes were blocked for 1 h in 5% BSA in PBS and incubated with primary and secondary antibodies (Supplementary Tables 2 and 3) for 1-2 h at room temperature, or at 4 °C overnight. Membranes were visualized and analyzed with the Odyssey CLx Infrared Imaging System (LI-COR Biosciences).

### Cell fixation and immunofluorescence
To image fluorescence, cells grown on coverslips were fixed for 15 min in 2% paraformaldehyde, incubated with DAPI for 5 min and mounted using Aqua-Poly/Mount (Polysciences, Inc.). For immunofluorescence, cells grown on coverslips were fixed in 2% paraformaldehyde and

permeabilized for 15 min in 0.1% Triton in PBS. To visualize LUD with CPD antibody, DNA was denatured for 5 min in 70 mM NaOH. Blocking was performed for 1 h in 3% BSA and 2.25% glycine in 0.05% Tween-PBS, followed by 2 h incubation at room temperature with primary antibodies and 1 h incubation with secondary antibodies (Supplementary Tables 2 and 3) and DAPI (Sigma-Aldrich). Coverslips were mounted on slides using Aqua-Poly/Mount (Polysciences, Inc.). To acquire images an LSM700 microscope equipped with 40x Plan-apochromat 1.3 NA oil immersion lens (Carl Zeiss Micro Imaging Inc.).

### Recovery of RNA synthesis
To measure recovery of RNA synthesis, cells were grown on coverslips and mock treated or irradiated with 2 J/m² UV-C (254 nm lamp, Philips). 24 h after irradiation, RNA was labeled by incubation with EU for 1 h. Cells were fixed in 4% paraformaldehyde and permeabilized with 0.1% Triton X-100 in PBS. Cells were incubated in Click-it buffer containing Atto 594 Azide (60 μM, Atto Tec.), Tris-HCl (50 mM, pH 7.6), CuSO4.5H2O (4 mM, Sigma) and ascorbic acid (10 mM, Sigma) for 1 h to visualize EU incorporation. DAPI (Sigma) was used to stain DNA and coverslips were mounted using Aqua- Poly/Mount (Polysciences, Inc.). Images were acquired on an LSM700 confocal microscope and transcription levels were quantified with FIJI image analysis software.

### Unscheduled DNA synthesis
To measure NER capacity by Unscheduled DNA synthesis[45], U2OS cells grown on coverslips were incubated with medium containing 0.5% FCS and 1 μM Palbociclib (Selleck Chemicals) for 24 h, to limit S phase cells. Subsequently, cells were treated with 16 J/m² UV irradiation and incubated for 1 h in medium containing 5-ethynyl-2′-deoxyuridine (EdU, Invitrogen). Cells were fixed in 3.6% formaldehyde, permeabilized with 0.1% Triton X-100 in PBS for 10 min and incubated with 1.5% BSA in PBS for 10 min. EdU incorporation was visualized by incubating cells for 1 h at room temperature with Click-it reaction cocktail containing Atto 594 Azide (60 μM, Atto Tec.), Tris-HCl (50 mM, pH 7.6), CuSO4*5H2O (4 mM, Sigma) and ascorbic acid (10 mM, Sigma). After washes in 0.1% Triton X-100 in PBS, DNA was stained with DAPI (Sigma), and slides were mounted using Aqua-Poly/Mount (Polysciences, Inc.). Images were acquired using an LSM700 confocal microscope and fluorescence signals were quantified using FIJI image analysis software. Background fluorescence was subtracted and intensity levels were averaged and normalized to the fluorescence levels in control conditions.

### Senescence reporter assay
To evaluate the induction of senescence features, on day 1 BFP-Lamin B1/mScarlet-Lamin A/GFP-IL6 transgenic U2OS and XPG KO cells were seeded. To deplete XPA, cells were transfected with siXPA on day 2. To evaluate DNA damage-induced senescence, cells were exposed to 1 μM doxorubicin (Sigma) for 2 h on day 2. Cells from all conditions were fixed on day 4 and imaged using a Leica SP8 confocal microscope equipped with a HC PL APO 40X/1.30 OIL CS2 objective. For the analysis of confocal images, we developed an image analysis pipeline in Fiji software[111]. The images were split into separate channels, one channel for each color. Images of the lamin A-intensity were segmented to obtain regions of interest (ROIs). For each obtained ROI, the mean intensities of transgenic lamin A, B1 and GFP were determined and exported to excel files. GFP-positive and negative cells were identified and plotted with their corresponding lamin intensities using Graphpad Prism v5.01 (Graphpad software Inc.).

### *C. elegans* strains, culture and imaging
*C. elegans* was cultured according to standard methods on nematode growth media (NGM) agar plates seeded with *Escherichia coli* OP50. *C.*

*elegans* strains used are listed in Supplementary Table 4. *C. elegans xpf-1* point mutants D719A and R805W were generated using gRNAs with targeting sequence TTCCTTTCAATTGCAATGTT (D719A) or ATCCCAA AATGAGATGTGTT (R805W) and as homology directed repair template ssDNA oligomers with sequence GGTAGACATTTTTTAATTAT TTTTTTATAACAATTACCGAAGAAGTCTCAGATCGGTGCTTACATTCT CTCGCCGAATATCGCCATCGAGAGGAAAGCGTTGGACGATTTGACAC AGTCTTTG (D719A) or AACTCTGCGGAATTTGTCGGAGAAATCGTC CACACACACCCACATCTTCGGATTTGCCCAAATTAAAGAGCAAAAAA TTGA (R805W). sgRNA and oligomer were injected together with Cas9 (Integrated DNA technologies) in *xpf-1(emc57[xpf::GFP])* animals[63]. Mutations were verified by genotyping PCR and sequencing. All mutants were backcrossed against wild-type strain, which was Bristol N2. All assays were performed at 20 °C.

### *C. elegans* imaging and recovery of protein synthesis
For imaging of DNA damage recruitment (Fig. 4A), living *C. elegans* were mounted on 2% agar pads in M9 buffer containing 10 mM NaN3 (Sigma) and imaged on a Leica SP8 confocal microscope. For quantification of DNA damage recruitment (Fig. 4B), animals were fixed at the indicated time points in 3.6% formaldehyde in M9 buffer and recruitment was determined by measuring the fluorescence signal in bivalents in oocytes, normalized for background fluorescence, on a Zeiss LSM700 confocal microscope. For recovery of protein synthesis (RPS)[70], AID::GFP was depleted by culturing young adult animals in the presence of 100 μM auxin (3-indoleacetic acid; Sigma) for 2 h. Animals were mock-treated or irradiated with 120 J/m² UV-B (Philips TL-12 tubes, 40 W) and allowed to recover for 48 h on culture plates. AID::GFP fluorescence was imaged in body wall muscle cells in the heads of living animals on a Leica TCS SP8 microscope (LAS AF software, Leica) and in Fiji ImageJ.

### L1 larvae survival assay
L1 larvae UV survival assays were performed as described[65]. Briefly, eggs collected from adult *C. elegans* after hypochlorite treatment were plated onto five technical replicate agar plates per UV dose seeded with HT115 bacteria. After 16 h, L1 larvae were irradiated with the indicated dose of UV-B (Philips TL-12 tubes, 40 W) and allowed to recover for 48 h. For illudin S survival, eggs collected from hypochlorite treatment were hatched in 96-well plates containing per well 100 μL of S Basal medium (supplemented with HT115 bacteria, 1 M potassium citrate (pH 6), trace metals solution, 1 M CaCl2 and 1 M MgSO4), while gently shaking at room temperature. After 16 h, the indicated concentrations of illudin S (Toku-e) were added to the liquid cultures for 24 h after which animals were allowed to recover on culture plates containing HT115 bacteria for 72 h at 15 °C. Animals arrested at the L1/L2 stages and surviving animals that developed beyond the L2 stage were counted to determine the survival percentage.

**Dye filling assays.** Dye filling experiments were performed as described[63]. Synchronized adult *C. elegans* were irradiated with the indicated dose of UV-B (Philips TL-12 tubes, 40 W) or incubated with the indicated dose of illudin S (Toku-e) for 24 h in 100 μL of S Basal medium (supplemented with HT115 bacteria, 1 M potassium citrate (pH 6), trace metals solution, 1 M CaCl2 and 1 M MgSO4) and allowed to recover on culture plates for 72 h. To deplete AID-tagged GTF2H1, animals were grown in the presence of 1 mM auxin (3-indoleacetic acid, Sigma) for 24 or 72 h, as indicated. Animals were then washed and incubated in 10 μg/ml DiI (Molecular probes) dissolved in M9 buffer for 30 min. Next, animals were allowed to recover for 1 h on culture plates after which dye filling was scored using an Olympus SZX12 Stereo Microscope equipped with a U-RFL-T mercury lamp or imaged using a Zeiss LSM700 confocal microscope.

## Statistical analysis

Mean values and S.E.M. error bars are shown for each experiment. For the statistical significance analysis of immobile fractions, we applied a One-Way ANOVA with correction for multiple comparisons. For iFRAP assays a Non-linear regression analysis was performed. All analyses were performed using Graph Pad Prism version 9 for Windows (GraphPad Software, La Jolla California USA).

## Reporting summary

Further information on research design is available in the Nature Portfolio Reporting Summary linked to this article.

## Data availability

All data supporting the findings of this study are available within the paper and its Supplementary Information. Any other data are available from the corresponding author upon request. Source data are provided with this paper.

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

## Acknowledgements

We thank Gert van Cappellen and Gert-Jan Kremers of the Erasmus MC Optical Imaging Center for microscope support, and Isabel Sierra of the Hubrecht Institute for critically reading the manuscript. This work was supported by the Netherlands Organization for Scientific Research (ALWOP.494 to HL; 711.018.007 to HL; VI.C.182.025 to JAM), Erasmus MC (HDMA Grant to WV/HL), Worldwide Cancer Research (15–1274 to WV/HL), the Dutch Cancer Society (KWF 10506 to WV/JAM/HL), the European Research Council (advanced grant 340988-ERC-ID to WV), and the Oncode Institute (to WV/RK/JAM), which is partly financed by the Dutch Cancer Society. We thank the Josephine Nefkens Cancer Program for infrastructure support.

## Author contributions

A.M.V., C.D.M., C.R.S., D.H. and U.U.K. performed live cell imaging and other human cell line experiments. C.D.M., M.W. and K.L.T. generated *C. elegans* strains, performed dye filling, survival and other *C. elegans* experiments. K.L.T. generated DNA constructs. A.M.V., B.H. and M.M.P.K. performed senescence experiments. A.M.V., C.R.S and A.F.T. generated cell lines. A.M.V., C.D.M., C.R.S., R.K., J.A.M., W.V. and H.L. conceptualized ideas, designed experiments and analyzed data. A.M.V., C.D.M. and M.M.P.K. generated figures. A.M.V., C.R.S. and H.L. wrote the manuscript. All authors reviewed the manuscript.

## Competing interests

The authors declare no competing interests.
