## [Peer Review File · Nature Communications]

Persistent TFIIH binding to non-excised DNA damage causes cell and developmental failureREVIEWER COMMENTS

Reviewer #1 (Remarks to the Author):

Here the authors attempted to correlate different disease subtypes and severity observed in xeroderma pigmentosum (XP) and Cockayne syndrome (CS) with TFIIH binding and removal. With FRAP and iFRAP, the authors first demonstrated that the mobility of XPB, a component of TFIIH, was significantly reduced in XPG/XPF deleted cells, which were reversed by co-depletion of XPA. Using the approaches of knockout (KO)/rescue and RNA interference in mammalian cells or *C. elegans*, it was shown that the severe phenotype in XPG/XPF deletion could be partially rescued by co-depletion of XPA or TTDA. It was therefore concluded that the prolonged binding of TFIIH to damaged DNA could explain the severe phenotype such as senescence or neuron dysfunction. Additionally, the data may also explain the phenotypic difference between XPA and XPF/XPG deficiency. In all, although the results in large part support their claims, additional experiments and explanations are required to strengthen their conclusions.

1. Although the data have shown the correlation between persistent TFIIH binding and either senescence or neuronal dysfunction upon XPG or XPF KO, it is unclear that whether the persistent lesions played any role. It is peculiar that with additional deletion of XPA or TTDA, these phenotypes could be partially rescued, as one would expect that the presence of these lesions would most likely trigger DNA damage response, leading to senescence. To address this point, CPD staining and EdU incorporation could be performed to determine whether lesions have been somehow partially removed under these conditions or if repair synthesis has occurred.
2. Fig. 2J and 2K are quite puzzling and an explanation is needed. It appears that the authors have chosen to show the effect of XPG KO and XPA si with reduction of cells in Q1 rather than increase in the other quadrants. Furthermore, XPG KO seemed to cause significant increase in IL-6+ cells but these cells appeared mostly in Q1 (Lamin A-/Lamin B -). In Fig. 2K, the comparison (P value) between - and + siXPA in the XPG KO group was not shown.
3. In Fig. S3C, deletion of GTF-2H5 seemed to have no effect on dil dye filling in contrast to XPA deletion (Fig. 3C) which showed a dramatic reduction in dye filling. What would be the possible explanation if both represent the defects in TFIIH binding? Furthermore, GTF-2H5 deletion seemed to have a dramatic effect on L1 survival (Fig. S3D).
4. Similarly in Fig. 4B, deletion of CSB did not appear to impact on dil dye filling, which is inconsistent with the involvement of TC-NER.
5. It is interesting that the degree of TFIIH immobility correlated with the disease severity from the three XPF mutants P379S, R799W, and C236R. It would be nice to know how these three mutations affect the nuclease activity of XPF.

Reviewer #2 (Remarks to the Author):

The current study aims to elucidate the heterogeneity observed in NER syndromes. By utilizing the *C. elegans* model system and human cells, the authors demonstrate that both XPA and TTDA facilitate the stable binding of TFIIH to DNA damage, whereas ERCC1-XPF and XPG promote the dissociation of TFIIH from DNA. They also observe that prolonged binding of TFIIH to DNA damage in *Errc1/Xpf* mutant cells results in senescence in both human cells and worms. Overall, these findings validate previous discoveries made by the same research group. However, there are concerns regarding the novelty and significance of the findings. To exclude the involvement of other functions of the ERCC1-XPF complex, such as repair of DNA interstrand crosslinks and resolution of transcription-associated R-loops, further experiments are required to investigate the persistent binding of TFIIH and its variations in distinct NER mutants. Additionally, the dissimilarity in TFIIH binding to DNA damage between XPA and XPF does not account for the comparable phenotypes observed in patients with XPA and XPF mutations. Moreover, heterogeneity exists not only among patients with mutations in different NER

genes but also among patients with different mutations within the same gene. The authors fail to provide specificity of the findings to TFIIH and its interaction with DNA damage. It would be important to explore whether differential TFIIH stalling in XPA or TTDA cells, compared to ERCC1-XPF or XPG cells, also affects other factors such as RNAPII stalling. Conducting additional experiments involving the crossbreeding of Xpa^{-/-} mice with Ercc1^{-/-} or Xpg^{-/-} mice could lend support to the proposed mechanism by assessing whether it alleviates the severe phenotype observed in Ercc1^{-/-} or Xpg^{-/-} mice. Lastly, the unexpected absence of neuronal UV sensitivity in Csb-1 mutants, as reported by the authors, raises further questions.

Reviewer #3 (Remarks to the Author):

The manuscript by Muniesa-Vargas et al. provides evidence that persistent binding of the THIIH transcription/ polymerase-accessory-complex, caused by blockage of Nucleotide Excision Repair (NER) at the step of endonuclease cleavage/excision of the damaged strand correlates with toxicity and developmental failure, both in human tissue culture models as well as in the *C. elegans* in vivo model. Outing myself as not being into the ins and outs of the inner NER community (and this might be important for judging novelty), I consider this study as overall very well done, and important. Improvements need to be implemented.

The manuscript would benefit from streamlining the text. Some suggestions are provided below. I am not a native speaker, but careful editing by a native speaker will be important. Some experimentation would strongly strengthen the major conclusion. The most important one is the analysis of catalytically dead XPF and XPG mutants, at least in *C. elegans* where this can be very easily done. The discussion could be clearer and testing transcriptional effects in worms should be very much doable. Further below, please find my comments, aligned with the progression of the text. When important, comments and suggestions are indicated as 'important'.

By way of introduction, the NER pathway comes in two flavors, global genome repair (gNER) is active throughout the genome, and mutations are primarily associated with Xeroderma pigmentosa, a heritable disease characterized by extreme UV sensitivity and cancer predisposition. In contrast, transcription-coupled repair (tcNER) specifically deals with the repair of transcribed DNA, when RNA polymerase extension is blocked by UV-induced lesions. Patients who are specifically defective for tcNER (eg CSB) suffer from Cockayne syndrome (CS) characterized by severe growth failure, progressive neurodegeneration, and progeria. Mutants of genes needed for both NER pathways, with phenotypes biased towards CS exist and these are characterized by the combined feature of XP and CS, (XPCS). tcNER requires the general multi-subunit transcription factor TFIIH. I understand that defective transcription has been linked to TFIIH instability, and that NER repair complexes have been reported to be continuously targeted to DNA damage in cells carrying and XPCS causative XPF-1 mutants, leading to the postulate that such accumulation may block transcription.

The authors show that XPA and TTDA general NER factors, XPCS causative mutants, as well as ERCC1/XPF and XPG nuclease defective mutants lead to TFIIH hyperaccumulation at DNA damage sites and that this is correlated with features of senescence in tissue culture cells and neurodegeneration in the *C. elegans* model.

List of comments

I found the paper at times very-very hard to read and provided suggestions for improvement further below. I think it is important to in Figure 1 always indicate if experiments were done immediately after UV irradiation and, and/or 3 hours after UV irradiation,

Lines 108. change to 'the fraction of immobilized TFIIH is decreased when FRAP experiments are conducted 3 hours after UV irradiation'.

Line 110. It is not entirely clear what 'this' refers to. 'These experiments'?

Line 111 'In sharp contrast, after XPA depletion this UV-dependent TFIIH immobile...' change to 'In sharp contrast, after XPA depletion the UV-dependent TFIIH immobile...'

Line 115 '..DNA incision by XPF and XPG allows TFIIH dissociation'... change to '..XPF and XPG allow TFIIH dissociation'...(((at this stage, you do not know if it is XPF and/or XPG-mediated incision is necessary for TFIIH dissociation. Eg says that 'the dissociation requires XPF or XPG'.

Line 119 'We therefore generated cells expressing fluorescent TFIIH with full XPA or XPF knockout (KO) to confirm our results and to test whether the strong UV-induced immobilization of TFIIH after XPF depletion can be suppressed by XPA loss.' Change to 'We, therefore, generated XPA or XPF knockout alleles in cells expressing fluorescent TFIIH to corroborate our RNAi-based results and to test if the strong UV-induced retention of TFIIH in UV treated XPF deficient cells is XPA dependent.'

Line 123 change to 'XPB locus in wild-type U2OS cells and in XPF KO U2OS cells we had previously generated'

Line 126, 'Moreover, 3 h after UV irradiation' change to 'Moreover, also 3 h after UV irradiation...'

Lines 98-135 in Figure 1 always indicate (and not just in A and B) that measurements were taken immediately after UV irradiation.

IMPORTANT: Figure 1F. Would you not need to also directly compare and indicate statistical information when comparing GFP-XPB immobilization in WT versus XPA deficient lines?

Line 128. Exchange 'more' with 'increased'.

IMPORTANT: Line 146. Can the authors make clear in the narrative that both lines carry the GFP knockin, one line carrying the XPF knockout, and both lines are otherwise isogenic (eg U2OS).

Line 177. My feeling is that Figure S1H is important and should be integrated into the main Figure.

Line 183. It is not exactly clear what 'which' refers to

Line 187. The connection between lamin (mis?) expression and senescence is not explained.

IMPORTANT: Line 190. Can a reference for doxorubicin treatment-mediated senescence induction be provided?

IMPORTANT: Line 187-197 Please improve the narrative to better describe the data, and better guide the reader. What do you mean by clear changes in lamin B1/A expression? The ratio of lamin B1 and A expression. For IL6 and lamin reporters clarify that the former is a transcriptional reporter and that the lamin reporters are translational reporters.

201 change to "...GFP fused to the TFIIH..."

IMPORTANT: Line 205. Can GFP-2H1/p62 be quantified?

Line 254 change 'mildly but significantly' to 'partially'

IMPORTANT: I suggest generating nuclease dead XPF-1 and XPG-1 mutants in *C. elegans* and possibly (to my mind this is not essential) in human cells. Only this way it can be clearly shown that a NER repair intermediate (eg persistent bubble) is required for excessive TFIIH retention.

IMPORTANT: Based on the results shown it appears most likely that the TFIIH retention at damage sites might be the toxic entity causing CSB-related phenotypes. Is it possible that this is due to a general reduction in transcription linked to the reduced availability of TFIIH for general transcription? What do we know about the stoichiometry of DNA damage sites and TFIIH complexes engaged in transcription? I understand that, using *C. elegans* the authors previously showed reduced transcription. Would it be possible to do this in this study as well, directly comparing the most important strains? Ideally, if the author's hypothesis is correct, which I consider likely, would it make sense to treat WT and various mutants with a low dose of UV which by itself does not cause a degenerative phenotype or reduced survival, and do this in conjunction with partial (RNAi mediated or degron mediated) TFIIH depletion which by in itself does not have an effect, but in conjunction with UV might lead to a synthetic effect? Are mutants of TFIIH subunits specifically defective for engaging with NER but not compromised for transcription known, and if so, could they be used to substantiate the claim that excessive TFIIH retention might be necessary for CSB-like pathology (eg compromised transcription).

REVIEWER COMMENTS

Reviewer #1 (Remarks to the Author):

*Here the authors attempted to correlate different disease subtypes and severity observed in xeroderma pigmentosum (XP) and Cockayne syndrome (CS) with TFIIH binding and removal. With FRAP and iFRAP, the authors first demonstrated that the mobility of XPB, a component of TFIIH, was significantly reduced in XPG/XPF deleted cells, which were reversed by co-depletion of XPA. Using the approaches of knockout (KO)/rescue and RNA interference in mammalian cells or *C. elegans*, it was shown that the severe phenotype in XPG/XPF deletion could be partially rescued by co-depletion of XPA or TTDA. It was therefore concluded that the prolonged binding of TFIIH to damaged DNA could explain the severe phenotype such as senescence or neuron dysfunction. Additionally, the data may also explain the phenotypic difference between XPA and XPF/XPG deficiency. In all, although the results in large part support their claims, additional experiments and explanations are required to strengthen their conclusions.*

We thank the reviewer for the constructive suggestions to strengthen our manuscript, which we have addressed as explained below.

1. Although the data have shown the correlation between persistent TFIIH binding and either senescence or neuronal dysfunction upon XPG or XPF KO, it is unclear that whether the persistent lesions played any role. It is peculiar that with additional deletion of XPA or TTDA, these phenotypes could be partially rescued, as one would expect that the presence of these lesions would most likely trigger DNA damage response, leading to senescence. To address this point, CPD staining and EdU incorporation could be performed to determine whether lesions have been somehow partially removed under these conditions or if repair synthesis has occurred.

We agree that it is useful to show whether DNA repair itself is equally affected by loss of XPA, XPF or XPG. Therefore, we performed Unscheduled DNA Synthesis experiments (EdU incorporation) in cells knockout for XPA, XPF, XPG or double knockout for XPA and XPF. These experiments are depicted in Figure S1H of the revised manuscript and show that NER is equally and completely abrogated in all knockout cells. In addition, it is important to note that we do not dismiss that persistent DNA damage by itself also negatively affects cell functionality. Some of the phenotypes that we study, such as development and dye filling in *C. elegans*, are clearly also observed in e.g. *xpa-1* deficient animals, but are more pronounced in *xpf-1* and *xpg-1* animals. That is why we indicate (in the discussion) that we hypothesize, based on our findings, that ‘persistent binding of TFIIH to DNA damage could be one of the causative factors contributing to a more severe pathology’.

2. Fig. 2J and 2K are quite puzzling and an explanation is needed. It appears that the authors have chosen to show the effect of XPG KO and XPA si with reduction of cells in Q1 rather than increase in the other quadrants. Furthermore, XPG KO seemed to cause significant increase in IL-6+ cells but these cells appeared mostly in Q1 (Lamin A-/Lamin B-). In Fig. 2K, the comparison (Pvalue) between – and + siXPA in the XPG KO group was not shown.

We thank the reviewer for indicating that our description was not sufficiently clear. In the text of the revised manuscript, we have now better indicated that changes in lamin B1 and lamin A expression are visualized by plotting these in a graph and by plotting the quantification of cells in each quadrant. Also, we now explicitly indicate that in these graphs, the IL-6 positive cells are shown with a blue color. In addition, to be clearer, we now show the increase of cells in the other quadrants (Q2-4) in Fig 2K, and show the change of cells in each individual quadrant separately in supplementary Fig 2F-I. We also added the statistic comparison between – and + siXPA in the XPG KO cells, as indicated by the reviewer.

It is true that many of the XPG KO cells that are in Q1 (already) show IL-6 expression. This is also observed after doxorubicin treatment, and in other senescence-related cellular models in which this reporter system is used and tested. This shows that there is no clear one-to-one correlation between lamin B1/lamin A expression ratio and IL-6 induction, which is true for many biomarkers of senescence. This data is not included in this manuscript, but will be part of a future publication describing this reporter system in more detail.

3. In Fig. S3C, deletion of GTF-2H5 seemed to have no effect on dil dye filling in contrast to XPA deletion (Fig. 3C) which showed a dramatic reduction in dye filling. What would be the possible explanation if both represent the defects in TFIIH binding? Furthermore, GTF-2H5 deletion seemed to have a dramatic effect on L1 survival (Fig. S3D).

It is true that *gtf-2H5* mutants are UV sensitive in the L1 larvae survival assay, but not so much in the dye filling assay. We do not know the exact reason for this, but *gtf-2H5* and *xpa-1* animals are not similar with regards to defects in TFIIH binding. XPA-1 is needed to promote stable binding to DNA damage and (helicase) activity of TFIIH. GTF-2H5 not only promotes this stable binding, but additionally promotes the stability of the whole TFIIH complex^{1,2} and the recruitment of XPA-1³. Stable TFIIH binding is therefore probably even more prevented in *gtf-2H5* animals than in *xpa-1* animals, and, possibly, therefore these animals are also less impaired in the dye filling assay than *xpa-1* animals. An alternative explanation may be that repair is not completely abrogated in *gtf-2H5* animals, and a low residual repair over the course of the dye filling experiment (which is three days) is sufficient to prevent dye filling defects. This type of residual repair is the likely explanation for the lack of a dye filling phenotype in *csb-1* animals, as explained below. For *csb-1* animals there is clear evidence for residual repair (via XPC-1), but for *gtf-2H5* animals there is no evidence for this. In contrast to dye filling, the L1 larvae UV survival not only reflects neuronal integrity (for which it can be used as readout), but also that of other cell types, such as glial and hypodermal cells, as we describe in the discussion of one of our previous papers⁴. Therefore, it could also simply be that this assay is a more sensitive readout of defects in NER and that therefore *gtf-2H5* mutants show UV sensitivity. It would be interesting to investigate this difference between *xpa-1* and *gtf-2H5* animals further, but as we do not think that these speculations add much to the conclusion of our current manuscript, we have not included these.

4. Similarly in Fig. 4B, deletion of CSB did not appear to impact on dil dye filling, which is inconsistent with the involvement of TC-NER.

We understand the concern of the reviewer, as it may seem illogical that CSB-1 deletion does not impact dye filling. First of all, it should be noted that in an *xpf-1* background, additional depletion of CSB-1 does partially suppress the dye filling defect (Figure 5B), indicating involvement of TC-NER via CSB-1. However, indeed the single *csb-1* mutant does not show a dye filling defect, which may be unexpected. The explanation that we provide for this is based on previous genetic analysis, which shows that in *C. elegans* XPC-1 via GG-NER can compensate and repair DNA damage, at least in part, in active genes in the absence of CSB-1^{4,5}. We confirm this also in the current manuscript by performing Recovery of Protein Synthesis assays in *csb-1* and *xpc-1* single and double mutants (Figure S3H), which is an indirect assay to measure repair in a transcribed gene. As this shows that there is residual repair (via XPC-1) in *csb-1* animals, this residual repair is likely sufficient over a period of 72 h in the dye filling assay to prevent a dye filling defect. Therefore, one should consider the impact of *csb-1* deletion in an *xpc-1* background, and indeed the double *xpc-1; csb-1* mutant has a strong dye filling defect (Figure 5C). To address the reviewer's concern, in the revised manuscript we have now explicitly stated that it is counterintuitive that *csb-1* animals do not show a dye filling defect, but that this is likely due to residual repair via XPC-1.

5. It is interesting that the degree of TFIIH immobility correlated with the disease severity from the three XPF mutants P379S, R799W, and C236R. It would be nice to know how these three mutations affect the nuclease activity of XPF.

We agree that it is relevant to discuss how XPF catalytic activity is affected by these mutations. Previous in vitro and cellular work has shown that the C236R mutation severely impairs XPF nuclease and repair activity, whereas with R799W and P379S mutations still residual nuclease and/or repair activity is observed⁶⁻⁹. In the revised manuscript, we now mention this in the results section and describe this in more detail in relation to our results in the discussion section.

Reviewer #2 (Remarks to the Author):

The current study aims to elucidate the heterogeneity observed in NER syndromes. By utilizing the C. elegans model system and human cells, the authors demonstrate that both XPA and TTDA facilitate the stable binding of TFIIH to DNA damage, whereas ERCC1-XPF and XPG promote the dissociation of TFIIH from DNA. They also observe that prolonged binding of TFIIH to DNA damage in Errc1/Xpf mutant cells results in senescence in both human cells and worms. Overall, these findings validate previous discoveries made by the same research group. However, there are concerns regarding the novelty and significance of the findings. To exclude the involvement of other functions of the ERCC1-XPF complex, such as repair of DNA interstrand crosslinks and resolution of transcription-associated R-loops, further experiments are required to investigate the persistent binding of TFIIH and its variations in distinct NER mutants. Additionally, the dissimilarity in TFIIH binding to DNA damage between XPA and XPF does not account for the comparable phenotypes observed in patients with XPA and XPF mutations.

We thank the reviewer for his/her critical concerns. Our results are indeed in line with and build further upon our previous findings regarding persistent accumulation of NER factors in XPCS complex cells. However, the results presented in this current paper are all novel and highly significant, as they show that longer TFIIH binding to DNA damage is toxic for cells, in particular for neurons as shown in a multicellular model organism, and that this can be suppressed by preventing stable TFIIH binding to DNA damage. It is important to stress that we do not exclude the involvement of other functions of ERCC1-XPF (or XPG) as important for disease phenotypes, as the reviewer seems to suggest. We are fully aware that ERCC1-XPF has other functions as well, such as in interstrand crosslink repair. After all, one of our co-authors was among the first to show this¹⁰, we have studied this in the past¹¹ (and still do), and this was, of course, clearly demonstrated by others^{12,13}. We do not claim that the difference in TFIIH binding that we observed explains all phenotypic differences between XPA, XPF and XPG deficiency and think that the involvement of ERCC1-XPF (and XPG) in other genome maintenance mechanisms contributes to symptoms observed in patients as well. This has also been clearly demonstrated for patients with e.g. Fanconi anemia (for ERCC1-XPF^{7,14}). Therefore, we wrote in the discussion of the submitted version of our manuscript that the difference between XPA and ERCC1-XPF deficiency can be partially attributed to different activities of these proteins. We now understand that we should indicate this more explicitly. Therefore, in the revised version, we provide examples of other genome maintenance pathways in which ERCC1-XPF and XPG function, and name Fanconi anemia as example. This way, we explain more clearly that our results indicate that persistence of stable NER intermediates *'likely also contributes to the more severe symptoms generally observed in absence of ERCC1-XPF or XPG'*. To emphasize this, we added the words 'in part' to the last sentence of the abstract, such that it does not appear that we claim that TFIIH DNA damage binding completely explains all phenotypic differences between XPA, TTDA, XPF and XPG deficiencies. Because we do not exclude other functions of ERCC1-XPF (or XPG), it is not clear to us which 'further experiments' would be required to investigate the persistent binding of TFIIH to address this concern, and hope the reviewer will agree with our answer.

Moreover, heterogeneity exists not only among patients with mutations in different NER genes but also among patients with different mutations within the same gene. The authors fail to provide specificity of the findings to TFIIH and its interaction with DNA damage. It would be important to explore whether differential TFIIH stalling in XPA or TTDA cells, compared to ERCC1-XPF or XPG cells, also affects other factors such as RNAPII stalling.

It is true that heterogeneity exists among patients with mutations in the same NER gene, which we also clearly indicate in the introduction. To address this with regard to heterogeneity due to XPF mutations in relation to TFIIH stalling, we performed experiments in which we monitored TFIIH stalling

in cells expressing different XPF mutants associated with heterogeneous disease features. This showed a clear correlation between TFIIH stalling and severity of XPF disease (shown in Figure 2F-I). Whether this holds also true for other NER genes whose mutation can lead to different diseases and severity is not the focus of this (initial) study, but is under current investigation in our group.

The reviewer indicates that it would be important to also explore if other factors are affected, such as RNAPII stalling. We agree with this, as we are aware that similar models based on a difference in RNAPII processing/stalling exist to explain heterogeneity of diseases caused by NER genes, such as CSB and UVSSA¹⁵⁻¹⁸. To address this, we performed FRAP in cells expressing CSB-mClover in which XPA, XPF or XPG was depleted. We used UV-induced immobilization of CSB (as measured by FRAP) as readout, as this is a very sensitive indicator of Pol II stalling, as previously shown (Geijer 2021; Llerena Schiffmacher 2023). We found, however, no clear differences between control cells and cells in which XPA, XPF or XPG was depleted. These new data are now shown in Figure 3B-C of the revised manuscript.

Conducting additional experiments involving the crossbreeding of Xpa^{-/-} mice with Ercc1^{-/-} or Xpg^{-/-} mice could lend support to the proposed mechanism by assessing whether it alleviates the severe phenotype observed in Ercc1^{-/-} or Xpg^{-/-} mice.

We agree with the reviewer that it would be a very good idea to next perform similar double mutant experiments in NER deficient mouse models. We have made plans to do this, but considering the time and funding needed for this, we think that these experiments are beyond the scope of the current manuscript.

Lastly, the unexpected absence of neuronal UV sensitivity in Csb-1 mutants, as reported by the authors, raises further questions.

We understand that the *csb-1* phenotype in the dye filling assay may be unexpected and also reviewer 1 asked about this. However, please note that neurons in *csb-1* deficient animals are sensitive to UV irradiation, which is clear from the fact that *csb-1* animals are hypersensitive in L1 larvae UV survival assays, which depends on neuronal transcriptional activity, which was published in previous papers by us and others^{5,19-21}. Also, we previously showed that UV sensitivity of *xpf-1* mutants in this assay can be rescued by expression of XPF-1 specifically in neurons only, but that this depends on the presence of CSB-1. This shows that neurons are UV hypersensitive without CSB-1. However, this UV hypersensitivity is not (yet) clearly visible in the dye filling assay at the UV doses that we used, and only becomes visible with higher UV doses. These higher UV doses are, however, not practical in this assay because these will also kill the animals themselves. As explained to reviewer 1, the fact that at relatively lower UV doses no dye filling defect is observed, as in *xpa-1/xpf-1* animals, is likely due to residual repair activity via XPC-1 in *csb-1* mutants. Previous genetic analysis already showed that in *C. elegans csb-1* mutants, XPC-1 via GG-NER can partially compensate and repair DNA damage in active genes^{4,5}. In the current manuscript, we confirm this by indirectly measuring repair in a transcribed gene using Recovery of Protein Synthesis assays (Figure S3H). This shows that in *csb-1* animals, there is residual repair via XPC-1. It is therefore likely that in the dye filling assay, which takes 72 h, residual XPC-1 mediated repair is sufficient to prevent a dye filling defect. For this reason, the impact of CSB-1 loss should be considered in an *xpc-1* deficient background. We found that double loss of XPC-1 and CSB-1 leads to a profound dye filling defect (Figure 5C). To address the reviewer's concern, we have now explicitly stated in the revised manuscript that it is counterintuitive that *csb-1* animals do not show a dye filling defect, and explained that this is likely due to residual repair via XPC-1.

Reviewer #3 (Remarks to the Author):

The manuscript by Muniesa-Vargas et al. provides evidence that persistent binding of the THIIH transcription/polymerase-accessory-complex, caused by blockage of Nucleotide Excision Repair (NER) at the step of endonuclease cleavage/excision of the damaged strand correlates with toxicity and developmental failure, both in human tissue culture models as well as in the C. elegans in vivo model. Outing myself as not being into the ins and outs of the inner NER community (and this might be important for judging novelty), I consider this study as overall very well done, and important. Improvements need to be implemented.

The manuscript would benefit from streamlining the text. Some suggestions are provided below. I am not a native speaker, but careful editing by a native speaker will be important. Some experimentation would strongly strengthen the major conclusion. The most important one is the analysis of catalytically dead XPF and XPG mutants, at least in C. elegans where this can be very easily done. The discussion could be clearer and testing transcriptional effects in worms should be very much doable. Further below, please find my comments, aligned with the progression of the text. When important, comments and suggestions are indicated as 'important'.

By way of introduction, the NER pathway comes in two flavors, global genome repair (gNER) is active throughout the genome, and mutations are primarily associated with Xeroderma pigmentosa, a heritable disease characterized by extreme UV sensitivity and cancer predisposition. In contrast, transcription-coupled repair (tcNER) specifically deals with the repair of transcribed DNA, when RNA polymerase extension is blocked by UV-induced lesions. Patients who are specifically defective for tcNER (eg CSB) suffer from Cockayne syndrome (CS) characterized by severe growth failure, progressive neurodegeneration, and progeria. Mutants of genes needed for both NER pathways, with phenotypes biased towards CS exist and these are characterized by the combined feature of XP and CS, (XPCS). tcNER requires the general multi-subunit transcription factor TFIIH. I understand that defective transcription has been linked to TFIIH instability, and that NER repair complexes have been reported to be continuously targeted to DNA damage in cells carrying and XPCS causative XPF-1 mutants, leading to the postulate that such accumulation may block transcription.

The authors show that XPA and TTDA general NER factors, XPCS causative mutants, as well as ERCC1/XPF and XPG nuclease defective mutants lead to TFIIH hyperaccumulation at DNA damage sites and that this is correlated with features of senescence in tissue culture cells and neurodegeneration in the C. elegans model.

List of comments

I found the paper at times very-very hard to read and provided suggestions for improvement further below.

We thank the reviewer for the suggestions to improve the readability of our manuscript. As suggested, our revised manuscript was critically read by a native English-speaking scientist, who provided us with helpful suggestions to streamline the text. These suggestions have all been incorporated in the revised text.

I think it is important to in Figure 1 always indicate if experiments were done immediately after UV irradiation and, and/or 3 hours after UV irradiation.

In the revised manuscript, we have indicated above all graphs whether the experiment was done immediately and/or 3 h after UV.

Lines 108. change to 'the fraction of immobilized TFIIH is decreased when FRAP experiments are conducted 3 hours after UV irradiation'.

We have adjusted this sentence.

Line 110. It is not entirely clear what 'this' refers to. 'These experiments'?

'This' indeed refers to the experiments and results described in the sentence before. We have now specified this by writing 'These results'.

Line 111 'In sharp contrast, after XPA depletion this UV-dependent TFIIH immobile...' change to 'In sharp contrast, after XPA depletion the UV-dependent TFIIH immobile...'

We changed this sentence as suggested by the native speaker that critically read our manuscript.

Line 115 '..DNA incision by XPF and XPG allows TFIIH dissociation'... change to '..XPF and XPG allow TFIIH dissociation'...(((at this stage, you do not know if it is XPF and/or XPG-mediated incision is necessary for TFIIH dissociation. Eg says that 'the dissociation requires XPF or XPG'

We agree that here we do not formally show that incision is necessary to allow TFIIH dissociation. We therefore changed these words to: 'and that XPF and XPG stimulate TFIIH dissociation'.

Line 119 'We therefore generated cells expressing fluorescent TFIIH with full XPA or XPF knockout (KO) to confirm our results and to test whether the strong UV-induced immobilization of TFIIH after XPF depletion can be suppressed by XPA loss.' Change to "We, therefore, generated XPA or XPF knockout alleles in cells expressing fluorescent TFIIH to corroborate our RNAi-based results and to test if the strong UV-induced retention of TFIIH in UV treated XPF deficient cells is XPA dependent.'

We have changed this sentence according as the reviewer requested. However, we did not change it exactly as the reviewer indicated, as this would have been inaccurate. The reason for this is that we did not generate XPF knockout alleles in cells expressing fluorescent TFIIH, but generated fluorescent TFIIH alleles in (already existing) XPF KO cells. This was explained in the sentence following this sentence. The sentence is now changed to: *We, therefore, generated cells with XPA or XPF knockout (KO) alleles expressing fluorescent TFIIH to corroborate our RNAi-based results and to test if the strong UV-induced retention of TFIIH in UV-treated XPF deficient cells is XPA dependent.*

Line 123 change to 'XPB locus in wild-type U2OS cells and in XPF KO U2OS cells we had previously generated'

We adjusted this sentence by adding 'U2OS cells', and also moved the 'previously generated' before 'XPF KO U2OS cells', as suggested by the native speaker that read our manuscript.

Line 126, 'Moreover, 3 h after UV irradiation' change to 'Moreover, also 3 h after UV irradiation...'

We added the word 'also'.

Lines 98-135 in Figure 1 always indicate (and not just in A and B) that measurements were taken immediately after UV irradiation.

As requested, we have indicated above all graphs when measurements were taken.

IMPORTANT: Figure 1F. Would you not need to also directly compare and indicate statistical information when comparing GFP-XPB immobilization in WT versus XPA deficient lines?

In this figure, the statistical comparison between WT and XPA KO cells was (and still is) indicated for each condition. We are therefore unsure of what the reviewer meant. Possibly, the reviewer meant the statistical comparison between siCTRL and siXPF in WT and between siCTRL and sXPF in XPA KO cells, which we have therefore added to the figure.

Line 128. Exchange 'more' with 'increased'.

We have changed to word 'more' into 'increased'.

IMPORTANT: Line 146. Can the authors make clear in the narrative that both lines carry the GFP knockin, one line carrying the XPF knockout, and both lines are otherwise isogenic (eg U2OS).

In accord with the reviewer's request, we now write in the text that 'the GFP-XPB KI wild type U2OS cells and GFP-XPB KI XPF KO U2OS cells' were locally UV irradiated.

Line 177. My feeling is that Figure S1H is important and should be integrated into the main Figure.

We moved this graph to the main figures, which is now shown as Figure 3A.

Line 183. It is not exactly clear what 'which' refers to.

The word 'which' referred to what is described in the subsentences before, i.e. that endogenous DNA damage accumulation causes cellular senescence and expression of senescence-associated secretory phenotype factors. To make the text clearer, we now started a new sentence in which we write: 'These DNA damage-induced senescence features are hypothesized to contribute...etc'.

Line 187. The connection between lamin (mis?) expression and senescence is not explained.

We added references and changed the text to make it clearer that changes in nuclear lamins are used as biomarker for senescence induction.

IMPORTANT: Line 190. Can a reference for doxorubicin treatment-mediated senescence induction be provided?

We added a reference to a general review that described therapy-induced senescence, including by doxorubicin.

IMPORTANT: Line 187-197 Please improve the narrative to better describe the data, and better guide the reader. What do you mean by clear changes in lamin B1/A expression? The ratio of lamin B1 and A expression. For IL6 and lamin reporters clarify that the former is a transcriptional reporter and that the lamin reporters are translational reporters.

We thank the reviewer for indicating our description was not sufficiently clear. As the reviewer requested, we changed the text such that we better explain how the reporter system works, i.e. which are translational and which are transcriptional reporters, what we observed and how this is plotted in the graphs. We hope that with these changes, the text is now easier to read and understand.

201 change to "...GFP fused to the TFIIH..."

We added the word 'the'.

IMPORTANT: Line 205. Can GFP-2H1/p62 be quantified?

We have quantified the AID::GFP::GTF-2H1 accumulation at chromosomes/bivalents in the oocytes of the animals shown. This confirmed that *xpf-1* animals show persistent recruitment of TFIIH, which is suppressed by addition loss of *xpa-1*. These new data are now shown in Figure 4B of the revised manuscript.

Line 254 change 'mildly but significantly' to 'partially'

We changed the words to 'partially'.

IMPORTANT: I suggest generating nuclease dead XPF-1 and XPG-1 mutants in C. elegans and possibly (to my mind this is not essential) in human cells. Only this way it can be clearly shown that a NER repair intermediate (eg persistent bubble) is required for excessive TFIIH retention.

We agree with the reviewer that it is useful to know whether a persistent bubble NER intermediate is involved in TFIIH retention. Already our experiments with the human XPF mutants (shown in Figure 2) address this question, because the C236R XPF mutant has severely reduced catalytic activity and hardly any DNA repair activity, while the P379S and R799W mutants do still show this, as has been previously shown⁶⁻⁸. We previously also showed that C236R behaves similar as a catalytic dead XPF mutant when it comes to accumulation of NER intermediates⁸. These experiments therefore already indicate that the more persistent TFIIH stalling is due to lack of catalytic incision (which also happens when XPF or XPG are completely absent as in the knockout cells). We understand that we did not clearly describe this in our manuscript, but have now added information on the catalytic and DNA repair activity of these mutants in the results and discussion texts of the revised version (also in response to reviewer 1). In addition, we generated two new mutants in *C. elegans*. As requested by the reviewer, one of these mutants is a catalytically inactive (nuclease dead) *xpf-1* mutant (D719A). The other mutant (R805W) mimics human XP mutation R799W, which still has catalytic activity. We performed dye filling experiments, which showed that the nuclease dead D719A mutation causes a severe neuronal defect after UV irradiation, while the other XP mutation does not. These results are in line with our other results and are added in Figure 4H of the revised manuscript.

We also generated *xpg-1* catalytic mutants in *C. elegans*, but did this already for another project involving another PhD student not involved in this manuscript. We prefer not to add these mutants to this manuscript. The reason for this is that the situation for XPG is different than for XPF, and also more complex. If XPF is absent or XPF is catalytically inactive, incision by XPG will also not occur, and the NER bubble intermediate will remain intact. An XPF catalytic mutant therefore shows similar severe phenotypes as XPF KO⁸. If XPG is absent, incision by XPF and XPG will also not occur, and the NER bubble intermediate will remain intact. However, if XPG is catalytically inactive, but physically still present at the lesion, incision by XPF still occurs and partial DNA repair synthesis takes place, as has been shown previously²². This therefore represents a different situation than when XPG or XPF are absent (knocked out) and a persistently stalled TFIIH bubble complex exists. Indeed, catalytically inactive *C. elegans xpg-1* mutants still show intermediate dye filling phenotypes (data not shown). This is furthermore also reflected by XPG patient and mouse model phenotypes. Patients with inactivating mutations in XPG merely display XP phenotypes, whereas patients that carry truncating XPG mutations, leading to the absence of XPG (resembling knockout), show a severe XPCS complex phenotype²³. Similarly, mice with nuclease inactivating point mutations in XPG endonuclease activity are mostly normal, whereas mice lacking XPG completely show progressive growth retardation and die prematurely²³. For these reasons, we do not think that a nuclease dead XPG mutant can be used to show that a NER repair intermediate is required for TFIIH retention (while XPG knockout can). As the situation for catalytically inactive XPG is more complex, we think this should be investigated and described more thoroughly than would be feasible for the (readability of the) current manuscript. As

these are also part of a follow up project executed by another PhD student, we hope that the reviewer understands that for these reasons we do not include these *xpg-1* mutants in this current manuscript.

IMPORTANT: Based on the results shown it appears most likely that the TFIIH retention at damage sites might be the toxic entity causing CSB-related phenotypes. Is it possible that this is due to a general reduction in transcription linked to the reduced availability of TFIIH for general transcription? What do we know about the stoichiometry of DNA damage sites and TFIIH complexes engaged in transcription? I understand that, using C. elegans the authors previously showed reduced transcription. Would it be possible to do this in this study as well, directly comparing the most important strains?

As we argue in the discussion, the toxicity of TFIIH retention at damage sites might be due to reduced availability of TFIIH for transcription. However, also other options or combinations of different options are possible, such as active signaling to shut down transcription, direct blockage of the transcription machinery by stalled TFIIH, or the formation of DNA breaks that impede with transcription. The reviewer asks if it is possible to show reduced transcription in *C. elegans*. We indeed showed reduced transcription of some housekeeping genes previously with qPCR in *gft-2H5* mutants¹, which could be attempted in *xpa-1*, *xpg-1*, *xpf-1* single and double mutants. However, a better and more straightforward approach, with which we have ample expertise and which is also standard in the field, would be to determine overall transcription levels in human cells, which cannot be as easily done in *C. elegans*. Therefore, we performed exactly this suggested experiment in human cells as shown in Figure 3A. This shows that transcription is indeed more impeded in XPF deficient cells than in XPA deficient cells and that XPA loss can partially suppress this stronger impediment. We do not think that trying to repeat this experiment in *C. elegans*, which will be technically much more challenging, will lead to new insight. To test if direct stalling/blockage of the transcription machinery is involved, we additionally performed additional FRAP experiments on CSB in human cells, as this was also asked by reviewer 2. CSB binding to DNA damage, which can be measured by FRAP, is a sensitive indicator of DNA damage-induced RNA polymerase II stalling^{24,25}. These experiments are shown in Figure 3B, but did not suggest that there is a difference in RNA polymerase II stalling in cells deficient for XPA, XPF or XPG.

Ideally, if the author's hypothesis is correct, which I consider likely, would it make sense to treat WT and various mutants with a low dose of UV which by itself does not cause a degenerative phenotype or reduced survival, and do this in conjunction with partial (RNAi mediated or degron mediated) TFIIH depletion which by in itself does not have an effect, but in conjunction with UV might lead to a synthetic effect?

We tried this experiment, out of curiosity if it would work, by growing animals on RNAi food that partially depletes TFIIH subunit p44/GTF-2H2C. However, we think that interpreting the outcome of this experiment may be difficult. The result of a successful attempt of the experiment is shown below. One problem was that we observed that RNAi by itself already affected dye filling capacity of *xpf-1*

mutants (for unknown reasons), as with 10 J/m² this was already lowered on control RNAi food (once, we also observed this for wild type animals). It may appear that growing *xpf-1* animals on p44 RNAi slightly lowers their dye filling capacity, possibly due to lower TFIIH availability. However, the fact that RNAi by itself already affects this, makes this interpretation difficult. In addition, the difficulty with this interpretation is that by

depleting this TFIIH subunit, the whole TFIIH complex will become unstable and non-functional and will therefore also fail to bind efficiently to damage sites during NER. Thus, not only its availability for transcription is affected, but also its DNA damage binding during NER. Therefore, not only will there be less TFIIH available for transcription, but also less TFIIH retention in *xpf-1* mutants. Finally, even if this experiment would work and clearly show an additive sensitivity, this still would not directly prove the hypothesis that TFIIH stalling in *xpf-1* mutants lowers its availability for transcription. The experiment would only show that this could be a possibility. For these reasons, we have decided not to include this experiment in the current manuscript. We are, however, still following up on these findings to try (in the future) to find out exactly how TFIIH stalling causes cell toxicity.

Are mutants of TFIIH subunits specifically defective for engaging with NER but not compromised for transcription known, and if so, could they be used to substantiate the claim that excessive TFIIH retention might be necessary for CSB-like pathology (eg compromised transcription).

Yes, one such mutant is known, which is the *gtf-2H5* mutant that we also use in our manuscript in Figures S3B-D to substantiate that the idea that DNA damage-induced TFIIH retention is toxic to neurons. TTDA/GTF25H is the smallest subunit of TFIIH, which is needed for recruitment of TFIIH and repair in all cells tested, but only for transcription in cells in which limiting TFIIH concentrations exist, such as in terminally differentiated cells of the human body^{3,26}. We previously confirmed that in *C. elegans*, the loss-of-function *gtf-2H5* mutant is, under normal laboratory conditions, compromised for NER but not transcription¹.

References

1. Thijssen, K.L., van der Woude, M., Davó-Martínez, C., Dekkers, D.H.W., Sabatella, M., Demmers, J.A.A., Vermeulen, W., and Lans, H. (2021). *C. elegans* TFIIH subunit GTF-2H5/TTDA is a non-essential transcription factor indispensable for DNA repair. *Commun. Biol.* **4**, 1336. [10.1038/s42003-021-02875-8](https://doi.org/10.1038/s42003-021-02875-8).
2. Theil, A.F., Nonnekens, J., Steurer, B., Mari, P.-O.O., de Wit, J., Lemaitre, C., Martejijn, J.A., Raams, A., Maas, A., Vermeij, M., et al. (2013). Disruption of TTDA results in complete nucleotide excision repair deficiency and embryonic lethality. *PLoS Genet.* **9**, e1003431. [10.1371/journal.pgen.1003431](https://doi.org/10.1371/journal.pgen.1003431).
3. Coin, F., De Santis, L.P., Nardo, T., Zlobinskaya, O., Stefanini, M., and Egly, J.M. (2006). p8/TTD-A as a repair-specific TFIIH subunit. *Mol. Cell* **21**, 215–226. [10.1016/j.molcel.2005.10.024](https://doi.org/10.1016/j.molcel.2005.10.024).
4. Sabatella, M., Thijssen, K.L., Davó-Martínez, C., Vermeulen, W., and Lans, H. (2021). Tissue-Specific DNA Repair Activity of ERCC-1/XPF-1. *Cell Rep.* **34**, 108608. [10.1016/j.celrep.2020.108608](https://doi.org/10.1016/j.celrep.2020.108608).
5. Lans, H., Martejijn, J.A., Schumacher, B., Hoeijmakers, J.H.J., Jansen, G., and Vermeulen, W. (2010). Involvement of global genome repair, transcription coupled repair, and chromatin remodeling in UV DNA damage response changes during developm. *PLoS Genet.* **6**, 41. [10.1371/journal.pgen.1000941](https://doi.org/10.1371/journal.pgen.1000941).
6. Ahmad, A., Enzlin, J.H., Bhagwat, N.R., Wijgers, N., Raams, A., Appeldoorn, E., Theil, A.F., J Hoeijmakers, J.H., Vermeulen, W., J Jaspers, N.G., et al. (2010). Mislocalization of XPF-ERCC1 nuclease contributes to reduced DNA repair in XP-F patients. *PLoS Genet.* **6**, e1000871. [10.1371/journal.pgen.1000871](https://doi.org/10.1371/journal.pgen.1000871).
7. Kashiyama, K., Nakazawa, Y., Pilz, D.T., Guo, C., Shimada, M., Sasaki, K., Fawcett, H., Wing,

- J.F., Lewin, S.O., Carr, L., et al. (2013). Malfunction of nuclease ERCC1-XPF results in diverse clinical manifestations and causes Cockayne syndrome, xeroderma pigmentosum, and Fanconi anemia. *Am. J. Hum. Genet.* *92*, 807–819. 10.1016/j.ajhg.2013.04.007.
8. Sabatella, M., Theil, A.F., Ribeiro-Silva, C., Slyskova, J., Thijssen, K., Voskamp, C., Lans, H., and Vermeulen, W. (2018). Repair protein persistence at DNA lesions characterizes XPF defect with Cockayne syndrome features. *Nucleic Acids Res.* *46*, 9563–9577. 10.1093/nar/gky774.
 9. Klein Douwel, D., Hoogenboom, W.S., Boonen, R.A., and Knipscheer, P. (2017). Recruitment and positioning determine the specific role of the XPF-ERCC1 endonuclease in interstrand crosslink repair. *EMBO J.* *36*, 2034–2046. 10.15252/EMBJ.201695223.
 10. Niedernhofer, L.J., Odijk, H., Budzowska, M., van Drunen, E., Maas, A., Theil, A.F., de Wit, J., Jaspers, N.G.J., Beverloo, H.B., Hoeijmakers, J.H.J., et al. (2004). The Structure-Specific Endonuclease Ercc1-Xpf Is Required To Resolve DNA Interstrand Cross-Link-Induced Double-Strand Breaks. *Mol. Cell. Biol.* *24*, 5776–5787. 10.1128/mcb.24.13.5776-5787.2004.
 11. Sabatella, M., Pines, A., Slyskova, J., Vermeulen, W., and Lans, H. (2020). ERCC1–XPF targeting to psoralen–DNA crosslinks depends on XPA and FANCD2. *Cell. Mol. Life Sci.* *77*, 2005–2016. 10.1007/s00018-019-03264-5.
 12. Klein Douwel, D., Boonen, R.A.C.M., Long, D.T., Szypowska, A.A., Räschle, M., Walter, J.C., and Knipscheer, P. (2014). XPF-ERCC1 Acts in Unhooking DNA Interstrand Crosslinks in Cooperation with FANCD2 and FANCP/SLX4. *Mol. Cell* *54*, 460–471. 10.1016/j.molcel.2014.03.015.
 13. Bhagwat, N., Olsen, A.L., Wang, A.T., Hanada, K., Stuckert, P., Kanaar, R., D’Andrea, A., Niedernhofer, L.J., and McHugh, P.J. (2009). XPF-ERCC1 Participates in the Fanconi Anemia Pathway of Cross-Link Repair. *Mol. Cell. Biol.* *29*, 6427–6437. 10.1128/mcb.00086-09.
 14. Bogliolo, M., Schuster, B., Stoepker, C., Derkunt, B., Su, Y., Raams, A., Trujillo, J.P., Minguillón, J., Ramírez, M.J., Pujol, R., et al. (2013). Mutations in ERCC4, encoding the DNA-repair endonuclease XPF, cause Fanconi anemia. *Am. J. Hum. Genet.* *92*, 800–806. 10.1016/j.ajhg.2013.04.002.
 15. Nakazawa, Y., Sasaki, K., Mitsutake, N., Matsuse, M., Shimada, M., Nardo, T., Takahashi, Y., Ohyama, K., Ito, K., Mishima, H., et al. (2012). Mutations in UVSSA cause UV-sensitive syndrome and impair RNA polymerase II processing in transcription-coupled nucleotide-excision repair. *Nat. Genet.* *44*, 586–592. 10.1038/ng.2229.
 16. Lans, H., Hoeijmakers, J.H.J., Vermeulen, W., and Marteijn, J.A. (2019). The DNA damage response to transcription stress. *Nat. Rev. Mol. Cell Biol.* *20*, 766–784. 10.1038/s41580-019-0169-4.
 17. Jia, N., Guo, C., Nakazawa, Y., van den Heuvel, D., Luijsterburg, M.S., and Ogi, T. (2021). Dealing with transcription-blocking DNA damage: Repair mechanisms, RNA polymerase II processing and human disorders. *DNA Repair (Amst)*. *106*, 103192. 10.1016/j.dnarep.2021.103192.
 18. Marteijn, J.A., Lans, H., Vermeulen, W., and Hoeijmakers, J.H.J. (2014). Understanding nucleotide excision repair and its roles in cancer and ageing. *Nat. Rev. Mol. Cell Biol.* *15*, 465–481. 10.1038/nrm3822.
 19. Babu, V., Hofmann, K., and Schumacher, B. (2014). A *C. elegans* homolog of the Cockayne syndrome complementation group A gene. *DNA Repair (Amst)*. *24*, 57–62. 10.1016/j.dnarep.2014.09.011.

20. Babu, V., and Schumacher, B. (2016). A *C. elegans* homolog for the UV-hypersensitivity syndrome disease gene UVSSA. *DNA Repair (Amst)*. *41*, 8–15. 10.1016/j.dnarep.2016.03.008.
21. van der Woude, M., and Lans, H. (2021). *C. elegans* survival assays to discern global and transcription-coupled nucleotide excision repair. *STAR Protoc*. *2*, 100586. 10.1016/j.xpro.2021.100586.
22. Staresincic, L., Fagbemi, A.F., Enzlin, J.H., Gourdin, A.M., Wijgers, N., Dunand-Sauthier, I., Giglia-Mari, G., Clarkson, S.G., Vermeulen, W., and Schärer, O.D. (2009). Coordination of dual incision and repair synthesis in human nucleotide excision repair. *EMBO J*. *28*, 1111–1120. 10.1038/emboj.2009.49.
23. Muniesa-Vargas, A., Arjan, , Theil, F., Ribeiro-Silva, C., Vermeulen, W., Lans, H., Theil, A.F., Ribeiro-Silva, C., Vermeulen, W., and Lans, H. (2022). XPG: a multitasking genome caretaker. *Cell. Mol. Life Sci*. 2022 793 79, 1–20. 10.1007/S00018-022-04194-5.
24. Geijer, M.E., Zhou, D., Selvam, K., Steurer, B., Mukherjee, C., Evers, B., Cugusi, S., van Toorn, M., van der Woude, M., Janssens, R.C., et al. (2021). Elongation factor ELOF1 drives transcription-coupled repair and prevents genome instability. *Nat Cell Biol* *23*, 608–619. 10.1038/s41556-021-00692-z [pii]10.1038/s41556-021-00692-z.
25. Llerena Schiffmacher, D.A., Kliza, K.W., Theil, A.F., Kremers, G.J., Demmers, J.A.A., Ogi, T., Vermeulen, M., Vermeulen, W., and Pines, A. (2023). Live cell transcription-coupled nucleotide excision repair dynamics revisited. *DNA Repair (Amst)*. *130*, 103566. 10.1016/j.dnarep.2023.103566.
26. Theil, A.F., Hoeijmakers, J.H.J., and Vermeulen, W. (2014). TTDA: big impact of a small protein. *Exp. Cell Res*. *329*, 61–68. 10.1016/J.YEXCR.2014.07.008.

REVIEWER COMMENTS

Reviewer #1 (Remarks to the Author):

The authors have properly addressed the concerns raised in previously review.

Reviewer #2 (Remarks to the Author):

I have carefully reviewed the revised manuscript and would like to express my gratitude for the authors' efforts in providing supportive evidence and further clarifications, which have greatly contributed to the overall quality of the paper. However, upon thorough consideration, I find it necessary to highlight a concern regarding the core findings of the study. Despite the valuable insights presented, there appears to be a lack of substantial evidence supporting the pivotal conclusion that knocking out XPA would alleviate the severity of "cellular impairment and NER disease." Unless I have misconstrued the proposed model, the interpretation of the findings seems to be in contrast to observed phenotypes, particularly in mice and certain patient cell lines documented in various sources, including publications and presentations by the Rotterdam group over the years. Specifically, my recollection suggests that the phenotype of double NER mutant animals, such as Xpc/Xpa, Xpd/Xpa, Csb/Xpa, and Csa/Xpa mice, is notably more severe compared to single NER mutant animals (Xpc, Xpd, Csb, and Csa). Considering the hypothesis that knocking out XPA could reduce the binding or stalling of TFIIH to DNA lesions, one would anticipate an "advantage" in double NER mutant animals with an XPA defective gene, resulting in an improved phenotype compared to their single mutant counterparts. To further substantiate the original hypothesis, I proposed that the authors consider conducting an experiment involving the crossbreeding of Ercc1 or Xpg mice (or any other NER mutant animal where transcription-blocking DNA lesions would accumulate) with Xpa mice. This would entail providing solid evidence demonstrating an improved disease phenotype, as hypothesized. I believe such an experiment is neither expensive nor difficult nor time-consuming, requiring only one breeding and a brief 1-month follow-up to observe any potential improvements in the phenotype of the double mutants compared to the single mutants. Regrettably, it appears that this suggested experiment was not conducted in the current study. I understand the constraints in research, but I strongly believe that the inclusion of this experiment is crucial for supporting the manuscript's conclusions. In the absence of solid evidence to the contrary, I am inclined to recommend that the manuscript not be accepted for publication in its present form. I appreciate the authors' dedication to advancing scientific knowledge and trust that my comments will be taken into consideration for the betterment of the manuscript.

Reviewer #3 (Remarks to the Author):

First, let me apologize for my late review. There was a miscommunication between the original NatureCom mail and my automated 'holiday response', which made me assume that I would not be called up for review after returning to work.

I overall support the manuscript. Also, extending into a mouse, double knockout studies, as another reviewer suggests, is far beyond the scope of the current manuscript.

While improving, the manuscript, especially part of the results section, needed to be better written. I am attaching edits using the word track program, which may help (file suggested edits). Please carefully go through those (I might not always be correct). I note that there the paper is co-authored by three veterans of the NER field. I suggest that the two spend an afternoon together to review the manuscript sentence by sentence.

As to science itself: Yes, the authors conducted experiments I suggested. Also, I understand why XPG

was omitted.

One thing is easily doable and relates to Figure S3H: Here, in all NER defective backgrounds, a near-complete absence of DNA repair synthesis is shown upon UV treatment. To argue that various repair defective strains are equally defective, I think a lower UV dose that only partially blocks repair DNA synthesis needs to be analyzed. If DNA repair synthesis is equally defective, the argument that primary repair defects and defects associated with decreased TFIIH2 dissociation are functionally distinct will become much stronger.

For materials and methods:

It is most important that all *C. elegans* names are given strain names unique to the author's lab and that these are shown in a table. The same holds true for any new alleles that were generated.

<https://wormbase.org/about/userguide/nomenclature#ik3b7ea5clm12064jfd89gh--10>

In the same line, I strongly suggest that all cell lines should be given a name, referring to a specific number in the author's collection.

REVIEWER COMMENTS

Reviewer #1 (Remarks to the Author):

The authors have properly addressed the concerns raised in previously review.

We again thank the reviewer for the constructive criticism.

Reviewer #2 (Remarks to the Author):

*I have carefully reviewed the revised manuscript and would like to express my gratitude for the authors' efforts in providing supportive evidence and further clarifications, which have greatly contributed to the overall quality of the paper. However, upon thorough consideration, I find it necessary to highlight a concern regarding the core findings of the study. Despite the valuable insights presented, there appears to be a lack of substantial evidence supporting the pivotal conclusion that knocking out XPA would alleviate the severity of "cellular impairment and NER disease." Unless I have misconstrued the proposed model, the interpretation of the findings seems to be in contrast to observed phenotypes, particularly in mice and certain patient cell lines documented in various sources, including publications and presentations by the Rotterdam group over the years. Specifically, my recollection suggests that the phenotype of double NER mutant animals, such as *Xpc/Xpa*, *Xpd/Xpa*, *Csb/Xpa*, and *Csa/Xpa* mice, is notably more severe compared to single NER mutant animals (*Xpc*, *Xpd*, *Csb*, and *Csa*). Considering the hypothesis that knocking out XPA could reduce the binding or stalling of TFIIH to DNA lesions, one would anticipate an "advantage" in double NER mutant animals with an XPA defective gene, resulting in an improved phenotype compared to their single mutant counterparts.*

We thank the reviewer for recognizing that our efforts have contributed to the quality of our paper. However, we do not agree that there is a lack of 'substantial evidence' supporting the conclusion that XPA loss alleviates the severity of phenotypes studied. We show this extensively and in multiple different ways, using different assays in both human cells and in *C. elegans*. We also do not agree that the interpretation of our findings is in contrast to observed phenotypes 'in mice and certain patient cell lines', as the reviewer suggests. The reviewer refers to publications and presentations by the 'Rotterdam group', by which we infer the group of Jan Hoeijmakers is meant. We would like to emphasize that this research group is different and independent from our research group.

The Hoeijmakers' group, and other groups, have indeed shown that *Xpa* knockout exacerbates the phenotype of *Csb*, *Csa* and *Xpd*(TFIIH) mutant mice. However, this is a very different situation compared to what we study in our paper. We did not study how XPA loss impacts *CSB*, *CSA* or *XPB*(TFIIH) mutant cells or *C. elegans*, but studied how this impacts *XPF* and *XPG* mutant cells and *C. elegans*. *XPF* and *XPG* have different functions and activities than *CSB*, *CSA* and *XPB* and our findings can therefore not be compared to these mutant mice. *XPF* and *XPG* act downstream of TFIIH and XPA, and are not necessary for TFIIH recruitment, but for its dissociation, as we have shown in our paper. Cells that lack *XPF* or *XPG* therefore have prolonged TFIIH binding to DNA damage, which can be (partially) suppressed by additional depletion of XPA. In contrast, *CSB* and *CSA* act upstream of TFIIH and *XPB* is part of TFIIH itself. *CSB*, *CSA* and *XPB* are therefore necessary for the recruitment of TFIIH (in TC-NER), and cells that lack these factors will therefore (probably) not have prolonged TFIIH binding to DNA damage. For this reason, it would not be logical to think that XPA depletion would reduce prolonged TFIIH binding in cells that have lost *CSB*, *CSA* or *XPB* function. We would therefore not 'anticipate an "advantage" in double NER mutant animals with an XPA defective gene' in *Csb*, *Csa* or *Xpd* animals, as the reviewer suggests, and our results certainly do not suggest this outcome for these mutants.

It is important to realize that in our *C. elegans* system we employ UV irradiation to enhance DNA damage levels, which is not employed in the mentioned mouse models to study developmental and neuronal problems. The reason for this is that *C. elegans* lives too short to accumulate sufficiently high endogenous DNA damage to manifest a phenotype. In these conditions, *xpf-1* and *xpg-1* *C. elegans* mutants have very severe neurological and developmental problems, while *xpa-1* mutants exhibit milder but still also clear and significant neurological and developmental problems. These phenotypes are reminiscent of the phenotypes observed in severe *XPF* and *XPG* deficient human patients, and also in less severe *XPA* patients. In contrast, *Xpa*, *Csb*, *Csa* and *Xpd* single mouse mutants have relatively mild neurological features. The mild mouse phenotypes are (at first sight) not consistent with the severe phenotypes observed in human patients, but we do not think that it is our responsibility, nor our focus, to try to explain this in our paper. We do, however, believe that it is worthwhile to discuss these findings in mice in our paper and have therefore described these in the discussion section of our revised manuscript.

To further substantiate the original hypothesis, I proposed that the authors consider conducting an experiment involving the crossbreeding of Ercc1 or Xpg mice (or any other NER mutant animal where transcription-blocking DNA lesions would accumulate) with Xpa mice. This would entail providing solid evidence demonstrating an improved disease phenotype, as hypothesized. I believe such an experiment is neither expensive nor difficult nor time-consuming, requiring only one breeding and a brief 1-month follow-up to observe any potential improvements in the phenotype of the double mutants compared to the single mutants. Regrettably, it appears that this suggested experiment was not conducted in the current study. I understand the constraints in research, but I strongly believe that the inclusion of this experiment is crucial for supporting the manuscript's conclusions. In the absence of solid evidence to the contrary, I am inclined to recommend that the manuscript not be accepted for publication in its present form. I appreciate the authors' dedication to advancing scientific knowledge and trust that my comments will be taken into consideration for the betterment of the manuscript.

We appreciate that the reviewer proposes valuable experiments using mouse models, and we already indicated in our previous rebuttal that such experiments would be a good follow up idea. However, we also indicated that considering the time and funding needed, these experiments are beyond the scope of our current paper. The reviewer indicates that such an experiment is not expensive or time-consuming, but we do not agree with this. We do not work with mice ourselves and will need to find a mouse group that is willing to collaborate with us. This will certainly require a significant amount of time and additional funding. According to Dutch (ethics) regulation, we would first need to write a study plan application, which will take at least 3 months to be approved. In the meantime, the collaborating lab could start crossing and breeding mice to generate double mutant mice, which will take (at least) two breeding rounds and not one as the reviewer writes. We expect that generating sufficient double mutant mice to reliably and reproducibly perform experiments will take at least a year. For these reasons, we still think that these mouse experiments are beyond the scope of our current paper. We hope the reviewer will appreciate that our manuscript highlights the value of a relatively simpler model like *C. elegans*, as a useful and ethically responsible alternative to the highly valued mouse models that are also used to study NER mechanisms and disease.

Reviewer #3 (Remarks to the Author):

First, let me apologize for my late review. There was a miscommunication between the original NatureCom mail and my automated 'holiday response', which made me assume that I would not be called up for review after returning to work.

I overall support the manuscript. Also, extending into a mouse, double knockout studies, as another reviewer suggests, is far beyond the scope of the current manuscript.

We thank the reviewer for the constructive comments and useful suggestions, which we have addressed as detailed below.

While improving, the manuscript, especially part of the results section, needed to be better written. I am attaching edits using the word track program, which may help (file suggested edits). Please carefully go through those (I might not always be correct). I note that there the paper is co-authored by three veterans of the NER field. I suggest that the two spend an afternoon together to review the manuscript sentence by sentence.

We thank the reviewer for many useful suggestions for clarifying our English text, many of which we have adopted. Although the manuscript was written by two first authors and the last author, we would like to stress that it was also thoroughly read and edited by the experienced senior scientists Roland Kanaar and Wim Vermeulen, who are also co-authors. Also, at the request of the reviewer, we asked an independent native English-speaking scientist (PhD) to read and copy-edit our manuscript. As a result, our manuscript had and still has a language quality score of 9 out of 10, as given by the Curie English editing service in the Research Square web environment associated with Springer Nature. We are therefore confident that our manuscript, especially in its current revised form, is sufficiently clear and well written.

As to science itself: Yes, the authors conducted experiments I suggested. Also, I understand why XPG was omitted.

One thing is easily doable and relates to Figure S3H: Here, in all NER defective backgrounds, a near-complete absence of DNA repair synthesis is shown upon UV treatment. To argue that various repair defective strains are equally defective, I think a lower UV dose that only partially blocks repair DNA synthesis needs to be analyzed. If DNA repair synthesis is equally defective, the argument that primary repair defects and defects associated with decreased TFIIH2 dissociation are functionally distinct will become much stronger.

We appreciate that the reviewer suggests an experiment to strengthen our conclusions. However, we think that the experiment shown in Figure S3H might be misinterpreted. Figure S3H shows the results of a 'Recovery of Protein Synthesis' assay, which was performed in wild type, *xpc-1*, *csb-1* single and *csb-1*; *xpc-1* double mutant animals. This experiment was included to substantiate the idea to the reader that in the absence of CSB-1 (=TC-NER), XPC-1-mediated NER is able to remove part of the DNA damage that is induced in actively transcribed genes of *C. elegans* postmitotic cells. We have shown this idea already previously using survival assays (refs 58 and 63; PMID 20463888; 33440146) and using a similar 'Recovery of Protein Synthesis' assay (ref 70; PMID 37522336). This assay measures the ability of cells to transcribe an *AID::GFP* transgene after DNA damage induction. Cells will only be able to do this, and transcribe genes after DNA damage induction, if the DNA damage in these genes is repaired. To indirectly measure gene transcription, GFP fluorescence levels are measured after GFP is first depleted using an AID degraon tag fused to GFP. Therefore, this assay is an indirect measure of DNA repair in transcribed genes, as we have extensively shown in our previous paper describing this method (ref 70; PMID 37522336).

The reviewer suggests to do this experiment with a lower UV dose to argue that various repair defective strains are equally defective. However, as explained above, this assay does not measure 'DNA repair synthesis', which is what the reviewer alludes to. Therefore, Figure S3H does not show '*a near-complete absence of DNA repair synthesis*' upon UV treatment, as the reviewer writes. Rather, the figure shows (as explained in the legends) the GFP protein levels in cells of 'untreated' animals, in cells of animals in which the GFP protein is 'depleted' (using an AID degraon tag fused to GFP) and in cells of UV-irradiated animals, following a recovery period of 48 h after depletion of the GFP protein (UV + recovery).

The reviewer suggests to use a lower UV dose '*to argue that various repair defective strains are equally defective*'. However, the *csb-1*, *xpc-1* and *csb-1;xpc-1* strains used in the assay shown in Figure S3H are not equally repair defective and we also do not claim or argue this in our manuscript. *csb-1* animals are defective in TC-NER and *xpc-1* animals are defective in GG-NER, and these animals are therefore not equally repair defective but each deficient in a different DNA repair subpathway. The double *xpc-1;csb-1* mutant is defective in both TC-NER and GG-NER, and therefore more repair defective than the *xpc-1* or *csb-1* single mutant, as also suggested by the assay shown in Fig. S3H.

The reviewer writes that UV '*partially blocks repair DNA synthesis*', which we find difficult to understand because UV irradiation does not block, but induces DNA repair-mediated DNA synthesis. To be able to directly measure this DNA repair synthesis, one should monitor the incorporation of novel nucleotides by the DNA repair machinery into the DNA. In the past, we have tried to set up such an assay in *C. elegans*, but we were never successful and I am not aware of a working protocol (in our lab) for this. This assay is, however, standard practice in human cultured cells. Indeed, to show that loss of XPA, XPF and XPG leads to similar DNA repair synthesis defects, we performed this 'Unscheduled DNA synthesis' in human cells. The results are shown in Figure S1H, which confirms that loss of XPA, XPF and XPG equally impair DNA repair.

For these reasons, we do not think that it makes sense to perform the assay shown in Figure S3H using a lower UV dose, as it will not show or prove what the reviewer asks. However, we understand that we may not have explained the assay shown in Figure S3H sufficiently well. Therefore, in the revised manuscript, this assay is now better explained.

For materials and methods:

It is most important that all C. elegans names are given strain names unique to the author's lab and that these are shown in a table. The same holds true for any new alleles that were generated. <https://wormbase.org/about/userguide/nomenclature#ik3b7ea5clm12064jfd89gh--10>

We are aware of the *C. elegans* nomenclature. Indeed, all new *C. elegans* strains and alleles are given names that refer to our lab (prefixes are, respectively, HAL and *emc*), which are also registered at Wormbase. All strains used are listed in Table S4.

In the same line, I strongly suggest that all cell lines should be given a name, referring to a specific number in the author's collection.

All cell lines used are listed in Table S1. It is not common in our (human) field to use a designated number or name for these newly generated cell lines, but all cell lines can be traced back in our collection of cells in our institute.